# Localization in the mapping particle filter

Juan M. Guerrieri[1,2,3], Manuel Pulido[1,3], Takemasa Miyoshi[4,5], Arata Amemiya[6], and Juan J. Ruiz[2,7,8]

[1]Departamento de Física, FaCENA, Universidad Nacional del Nordeste, Corrientes, Argentina
[2]Departamento de Ciencias de la Atmósfera y los Océanos, FCEyN, Universidad de Buenos Aires, Buenos Aires, Argentina
[3]IMIT, CONICET, Corrientes, Argentina
[4]RIKEN Center for Computational Science, Kobe, Japan
[5]RIKEN Center for Interdisciplinary Theoretical and Mathematical Sciences, Kobe, Japan
[6]Japan Weather Association, Tokyo, Japan
[7]Centro de Investigaciones del Mar y la Atmósfera, CIMA/CONICET-UBA, Buenos Aires, Argentina
[8]Instituto Franco-Argentino para el Estudio del Clima y sus Impactos (IRL IFAECI/CNRS-IRD-CONICET-UBA), Buenos Aires, Argentina

**Correspondence:** Juan M. Guerrieri (juanmguerrieri@comunidad.unne.edu.ar)

**Abstract.**

Data assimilation involves sequential inference in geophysical systems with nonlinear dynamics and observational operators. Non-parametric filters are a promising approach for data assimilation because they are able to represent non-Gaussian densities. The mapping particle filter is an iterative ensemble method that incorporates the Stein Variational Gradient Descent (SVGD) to produce a particle flow transforming state vectors from prior to posterior densities. At every pseudo-time step, the Kullback-Leibler divergence between the intermediate density and the target posterior is evaluated and minimized. However, for applications in geophysical systems, challenges persist in high dimensions, where sample covariance underestimation leads to filter divergence. This work proposes two localization methods, one in which a local kernel function is defined and the particle flow is global. The second method, given a localization radius, physically partitions the state vector and performs local mappings at each grid point. The performance of the proposed Local Mapping Particle Filters (LMPFs) is assessed in synthetic experiments. Observations are produced with a two-scale Lorenz system, while a one-scale Lorenz model is used as surrogate, introducing model error in the inference. The methods are evaluated with both full and partial observations, as well as with different linear and non-linear observational operators. The LMPFs with Gaussian mixtures in the prior density perform similarly to Gaussian filters such as the Ensemble Transform Kalman Filter (ETKF) and the Local Ensemble Transform Kalman Filter (LETKF) in most cases, and in some scenarios, they provide competitive performance in terms of analysis accuracy.

## 1 Introduction

Particle filters have emerged as a valuable approach for addressing non-linear data assimilation challenges, especially in the context of geophysical systems, with particular promise for improving short-term meteorological forecasting. This potential derives from the inherently non-Gaussian nature of convective instabilities—which dominate short-term weather patterns—and their rapid growth rates compared to synoptic-scale phenomena (Hohenegger and Schar, 2007). As model resolution increases and observation operators become more complex, including non-linear relationships with the model state, the challenge of

accurately representing these growing non-linear and non-Gaussian features becomes more pronounced. Gaussian data assimilation techniques, such as Kalman filter-based methods, encounter limitations when confronted with non-linearity. These methods assume a Gaussian prior probability density function for the state. Variational methods struggle under strong non-Gaussianity resulting in multimodal cost functions or when the observational errors deviate from being Gaussian as well. Ensemble Kalman filters (EnKFs) explicitly assume that the prior density function and the observation likelihood follow a Gaussian distribution. Notably, Ruiz et al. (2021) show that even when drastically reducing the assimilation window of the Local Ensemble Transform Kalman Filter (LETKF), first introduced by Hunt et al. (2007), from 5 minutes to 30 seconds in 1 km-resolution experiments, residual non-Gaussianity persists at 40% levels.

In contrast, particle filters are non-parametric and offer distinct advantages in handling non-Gaussian error statistics (van Leeuwen et al., 2019). However, particle filters face challenges when dealing with high dimensionality, which is particularly prominent in geophysical applications characterized by a large number of variables. The standard Sequential Importance Resampling filter (SIR, Doucet et al., 2001) preserves and statistically replicates only the particles near observations, leading to sample impoverishment and weight degeneracy. To address this issue, a proposal density can incorporate information from both model dynamics and current observations, guiding particles toward high-probability regions and improving particle diversity by updating weights based on the ratio between the proposal density and the actual posterior density (van Leeuwen et al., 2019).

The problem of high dimensionality has also led to the development of several methods. Localization was first introduced for particle filters independently in Bengtsson et al. (2003) and van Leeuwen (2003). Other implementations can be found in Poterjoy (2016); Robert and Künsch (2017). Further methods are based on tempering (Neal, 1996), which mitigate the computational burden, instability and inaccuracy associated with high-dimensional problems, and jittering (Cotter el al., 2020), also referred to as regularisation, used to rejuvenate particles before or after resampling, as well as after tempering steps. An alternative approach to overcome these limitations is provided by particle flow filters (PFFs, Daum et al., 2010 and van Leeuwen et al., 2019). Instead of relying on the two-step process of weighting and resampling, PFFs move particles continuously through state space via a differential equation over a pseudo-time —drawing on ideas from Markov chain Monte Carlo (MCMC) methods such as those in Gallego and Ríos Insua (2018)— transforming the prior into the posterior distribution without modifying particle weights and thus avoiding the resampling and jittering steps required in traditional particle filters.

This work is concerned with developing a localization scheme for the variational mapping particle filter (MPF) proposed by Pulido and van Leeuwen (2019). The MPF is a particle flow filter that holds potential for non-linear applications in meteorology and oceanography. It is a sequential Monte Carlo algorithm that uses the Stein Variational Gradient Descent (SVGD) method, proposed by Liu and Wang (2016). In the MPF, state vectors, also known as particles, are propagated from the state predicted by the model (referred to as the background or forecast state) to states whose probability density function matches the posterior density, through a series of mappings. These gradient descent mappings aim to minimize the Kullback-Leibler divergence between the posterior density, which is obtained by applying Bayes' formula, and the sequence of intermediate densities.

The SVGD is a deterministic inference algorithm that converges in the limit of many particles (Del Moral, 2013), but it still faces the commonly referred problem known as 'the curse of dimensionality', for representing densities in high-dimensional

spaces. This is a common problem in particle filters (Snyder et al., 2008). One of its manifestations is the underestimation of the sample covariance and the subsequent divergence of the filter. Zhuo et al. (2018) demonstrated that SVGD often collapses into the modes of the target distribution, and this drawback becomes more severe with higher dimensions. Additionally, Ba et al. (2022) have demonstrated that SVGD-based algorithms offer few convergence guarantees. This issue persists even when the number of particles (or ensemble members) is larger than the dimension of the state. Among these limited cases, convergence is achievable in the mean-field regime, which occurs when the number of particles tends to infinity. To improve its convergence properties, Ba et al. (2022) proposed alternative formulations of SVGD. Furthermore, the SVGD produces biased samples for finite ensemble size and only produces unbiased samples for infinite ensemble size. Gallego and Ríos Insua (2018) solved the bias issue by introducing stochastic noise that makes the method unbiased for any ensemble size. Leviyev et al. (2022) provided an efficient methodology to incorporate the stochastic noise, and also accelerated the convergence through a Newton method that incorporates information from the Hessian. Finally, Ma et al. (2015) offered a complete and general framework for designing samplers based on the SVGD that guarantee a correct stationary distribution and facilitate the exploration of the space.

In the field of geophysical modeling, ensemble-based methods and particle filters are recognized as key frameworks for data assimilation. Both can incorporate localization methods to enhance their performance. Localization is a well-founded assumption considering that the state-dependent correlation between physical variables decreases with the distance between them. In the context of these frameworks, localization techniques serve the purpose of reducing the dimensionality of the assimilation process, ensuring accurate integration of observed data into the model state. For the EnKF, localization is typically achieved by adjusting the influence of observations and the prior error covariances based on their spatial proximity to the estimation point (Houtekamer and Mitchell, 2001; Hamill and Whitaker, 2001; Whitaker and Hamill, 2002). Developing and implementing these localization techniques within the EnKF and particle filters are critical for optimizing their effectiveness in real-world scenarios with spatio-temporal dynamics.

In particle filters, localization can be implemented in many ways (Farchi and Bocquet, 2018), including resampling-based approaches. For instance, Penny and Miyoshi (2016) proposed a local particle filter (LPF) that uses observation-space localization to compute independent analyses at each grid point. By applying deterministic resampling and smoothing the analysis weights across neighboring points, the LPF effectively mitigates particle degeneracy and enhances performance in highly nonlinear and non-Gaussian scenarios. While that work demonstrates the advantages of resampling-based localization, alternative particle flow-based methods avoid resampling and apply continuous transformations to particles from the prior density to the posterior.

Hu and van Leeuwen (2021) addressed the application of the Particle Flow Filter (algorithm based on what we call MPF) in high-dimensional systems, evaluating its performance in a Lorenz-96 system with 1,000 variables and 20 particles, observing 25% of the state variables and using three different observation operators. To avoid the problem of marginal distribution collapse in sparsely observed, high-dimensional settings, they proposed the use of a matrix-valued kernel, noting that a scalar kernel failed in these scenarios. They implemented a preconditioning matrix within the particle flow formulation to accelerate convergence. This matrix was chosen as the localized prior covariance matrix, which was localized using a Schur product

with a distance-decaying matrix. This approach resulted in the cancellation of the prior covariance matrix in the particle flow expression. Its performance was comparable to the LETKF and did not require explicit covariance inflation. This method was also applied to a full atmospheric model in Hu et al (2024).

In this work, two localization schemes in the MPF are introduced to reduce dimensionality and mitigate the problem of the curse of dimensionality in the MPF. These schemes are evaluated in the two-scale Lorenz model using both total and partial observations and nonlinear observation operators.

The work is structured as follows: section 2 introduces two LMPFs methodologies, section 3 describes the experimental design, section 4 presents the results of the experiments and section 5 draws the conclusions of the work.

## 2 Methodology

### 2.1 Mapping particle filter's review

The Mapping Particle Filter (MPF), introduced by Pulido and van Leeuwen (2019), is a non-parametric deterministic data assimilation method based on sample points, i.e., particles. It involves the transformation of the sample states from a prior density function to a posterior density by passing through intermediate states. These intermediate states are driven by an interacting particle flow designed to minimize the Kullback-Leibler divergence between a kernelized distribution of the sample states and the target posterior distribution. Subrahmanya et al. (2025) presents a general formulation to minimize the KL divergence. It formulates the flow field using the Fokker-Planck equation to evolve particles and sample the posterior distribution without using a reproducing kernel Hilbert space.

In the MPF, based on a hidden Markov model, a state vector evolves over time using a dynamical model and is observed using an observational model simultaneously,

$$\mathbf{x}_k = \mathcal{M}(\mathbf{x}_{k-1}, \boldsymbol{\eta}_k), \tag{1}$$

$$\mathbf{y}_k = \mathcal{H}(\mathbf{x}_k, \boldsymbol{\nu}_k), \tag{2}$$

where $\mathbf{x}_k \in \mathbb{R}^{N_x}$ represents the state at time $k$, $\mathcal{M}$ is the model operator, $\boldsymbol{\eta}_k$ denotes the random model error, $\mathbf{y}_k \in \mathbb{R}^{N_y}$ are the observations, $\mathcal{H}$ is the observation operator, and $\boldsymbol{\nu}_k$ represents the observational error. Here, a general framework is presented in which both model and observational errors can be non-additive.

The target density function of the particle flow corresponds to the posterior probability density using Bayes' formula during the assimilation stage,

$$p(\mathbf{x}_k|\mathbf{y}_{1:k}) = \frac{p(\mathbf{y}_k|\mathbf{x}_k)p(\mathbf{x}_k|\mathbf{y}_{1:k-1})}{p(\mathbf{y}_k|\mathbf{y}_{1:k-1})}, \tag{3}$$

This probability density function delineates the analysis states by capturing the likelihood of the forecast given a particular set of observations and a specified prior density.

Consider a set of $N_p$ particles $\{\mathbf{x}_{k-1}^{(1:N_p)}\}$ that samples the posterior density at time $k-1$. To obtain a state that matches the posterior density at time $k$, the MPF iteratively computes intermediate states from the prior to the target. The particles that

sample the prior density at time $k$ are states that undergo dynamical evolution from the particles that sample the posterior density at time $k-1$, denoted as $\{\mathbf{x}_k^{f(j)} = \mathbf{x}_{k,0}^{(j)} = \mathcal{M}(\mathbf{x}_{k-1}^{a(j)}, \boldsymbol{\eta}_k^{(j)})\}_{j=1}^{N_p}$, where the second subscript represents the pseudo-time of the mapping iteration. The superscript $f(j)$ indicates the $j$th particle of the forecasted states, and $a(j)$ indicates the $j$th particle of the analysis states (previous estimates). At each iteration, the particles are transformed by

$$\mathbf{x}_{k,i}^{(j)} = T(\mathbf{x}_{k,i-1}^{(j)}) = \mathbf{x}_{k,i-1}^{(j)} + \epsilon\, \mathbf{v}(\mathbf{x}_{k,i-1}^{(j)}), \tag{4}$$

where $T$ represents the iteration mapping, $\mathbf{v}$ represents the velocity of the particle flow in pseudo-time, $\epsilon$ represents the step size of the mapping. It may be considered fixed or estimated adaptively by means of stochastic optimization algorithms (Kingma and Ba, 2014).

The velocity seeks to minimize the Kullback-Leibler divergence between the target posterior density function, and the density of the intermediate states. Therefore, the sample from the prior density is transformed towards a sample from the posterior density through a set of discrete transformations, which in the infinitesimal limit may be interpreted as a flow in the state space.

MPF is inspired by the Stein Variational Gradient Descent method (Liu and Wang, 2016) which is kernel-based. These methods are algorithms that rely on kernel functions to measure similarities between state vectors from different particles. The MPF selects a space of functions known as the unit ball of a reproducing kernel Hilbert space (RKHS), denoted as $\mathbb{F}$. The optimization task is to find $\mathbf{v} \in \mathbb{F}$ that indicates the steepest descent direction of the Kullback-Leibler Divergence $D_{KL}$ between the target posterior density and the intermediate density.

By choosing an isotropic kernel $K$ and given a set of particles $\{\mathbf{x}_{k,i-1}^{(1:N_p)}\}$ representing a sample of the intermediate density at pseudotime $i-1$, the gradient of the Monte Carlo integration of the KL divergence is computed as:

$$\mathbf{v}(\mathbf{x}) = \frac{1}{N_p} \sum_{j=1}^{N_p} \left[ K(\mathbf{x}_{k,i-1}^{(j)}, \mathbf{x}) \left( \nabla_{\mathbf{x}_{k,i-1}^{(j)}} \log p(\mathbf{x}_{k,i-1}^{(j)}) + \nabla_{\mathbf{x}_{k,i-1}^{(j)}} \log K(\mathbf{x}_{k,i-1}^{(j)}, \mathbf{x}) \right) \right]. \tag{5}$$

The first term in the parenthesis of eq. (5), called the kernel-smoothed gradient of the posterior density, acts as a central force guiding the samples from an initial distribution density function towards the modes of the posterior density. The second term acts as the repulsive force and prevents the particles from collapsing into modes of the posterior. Note that the variables in eq. (4) and eq. (5) are nondimensionalized by proper scaling as in previous works (Pulido and van Leeuwen, 2019; Liu and Wang, 2016; Lu et al, 2019). To consider dimensional variables one may rewrite $\epsilon = D\delta t$, where $D$ is a diffusion coefficient and $\delta t$ is the pseudo-time step. In that case, the diffusion coefficient controls the optimization convergence rate as in gradient flows (Jordan et al., 1998) and must be incorporated into the velocity term. In this work, however, we keep $\epsilon$ as the single effective parameter controlling the convergence rate and adapt it during gradient descent using low-order momentum estimates (Kingma and Ba, 2014).

Radial basis functions are used as kernels in this work,

$$K(\mathbf{x}, \mathbf{x}') = e^{-\frac{1}{2}\|\mathbf{x}-\mathbf{x}'\|_\Sigma^2}, \tag{6}$$

where $\|\mathbf{x} - \mathbf{x}'\|_\Sigma^2 = (\mathbf{x} - \mathbf{x}')^\top \Sigma^{-1}(\mathbf{x} - \mathbf{x}')$ denotes the square of the Mahalanobis distance and $\Sigma$ is referred to as the kernel co-variance matrix. This matrix needs to be specified at the beginning of the process. In this work, it is assumed to be proportional to the forecast covariance matrix, though other approaches for defining it are possible.

The gradient of the logarithm of the posterior density requires the analytical forms of the prior density and the likelihood function. In this work, observational errors are assumed to be additive and Gaussian, but the framework is general and other observational error distributions may be considered. The resulting gradient of the log posterior density, evaluated at $\mathbf{x}_{k,i-1}^{(j)}$ is:

$$\nabla_{\mathbf{x}_{k,i-1}^{(j)}} \log p\left(\mathbf{x}_{k,i-1}^{(j)}\right) = \mathbf{H}^\top \mathbf{R}_k^{-1}\left(\mathbf{y}_k - \mathcal{H}\left(\mathbf{x}_{k,i-1}^{(j)}\right)\right) + \nabla_{\mathbf{x}_{k,i-1}^{(j)}} \log p(\mathbf{x}_{k,i-1}^{(j)}|\mathbf{y}_{1:k-1}). \tag{7}$$

The first term is the observation likelihood function, in which $\mathcal{H}$ is the observational operator, $(\in \mathbb{R}^{N_y \times N_x})$, and $\mathbf{H}$ denotes the tangent linear observation operator, $\mathbf{H} = \frac{d\mathcal{H}}{d\mathbf{x}}(\mathbf{x})$, while $\mathbf{R}$ stands for the observational error covariance matrix $(\in \mathbb{R}^{N_y \times N_y})$. The second term in eq. (7) is the gradient of the logarithm of the prior density.

In the case of a Gaussian prior density, where

$$p(\mathbf{x}_k^{(j)}|\mathbf{y}_{1:k-1}) = Z \cdot e^{-\frac{1}{2}\left\|\mathbf{x}_{k,i-1}^{(j)} - \overline{\mathbf{x}}_{k,0}\right\|_{\mathbf{B}_k}^2}, \tag{8}$$

the second term is reduced to

$$\nabla_{\mathbf{x}_{k,i-1}^{(j)}} \log p(\mathbf{x}_k^{(j)}|\mathbf{y}_{1:k-1}) = -\mathbf{B}_k^{-1}\left(\mathbf{x}_{k,i-1}^{(j)} - \overline{\mathbf{x}}_{k,0}\right), \tag{9}$$

where $Z$ is the normalizing constant, $\mathbf{B}_k$ is the prior or background covariance matrix and $\overline{\mathbf{x}}_{k,0}$ is the prior mean. For sequential Monte Carlo, the prior density in eq. (9) is given by the forecast density, such that $\mathbf{B}_k = \hat{\mathbf{P}}_k^f$.

Alternatively, if we assume the prior density is a Gaussian mixture based on the forecast particles, the prior is

$$p(\mathbf{x}_k^{(j)}|\mathbf{y}_{1:k-1}) = Z \cdot \exp\left\{-\frac{1}{2}\left\|\mathbf{x}_{k,i-1}^{(j)} - \boldsymbol{\mu}_{k,i,j}\right\|_{\mathbf{Q}_k}^2\right\}, \tag{10}$$

where $\boldsymbol{\mu}_{k,i,j} = \frac{\sum_{m=1}^{N_p} \psi_{kijm}\mathcal{M}(\mathbf{x}_{k-1}^{(m)})}{\sum_{m=1}^{N_p} \psi_{kijm}}$ are the Gaussian centroids, $\psi_{k,i,j,m} = \exp\left[-\frac{1}{2}\left\|\mathbf{x}_{k,i-1}^{(j)} - \mathcal{M}(\mathbf{x}_{k-1}^{(m)})\right\|_{\mathbf{Q}_k}^2\right]$ that represents the adaptive weights, and $\mathbf{Q}_k$ represents the covariance matrix of the Gaussian mixture. The second term in eq. (7) results in

$$\nabla_{\mathbf{x}_{k,i-1}^{(j)}} \log p(\mathbf{x}_k^{(j)}|\mathbf{y}_{1:k-1}) = -\mathbf{Q}_k^{-1}\left[\mathbf{x}_{k,i-1}^{(j)} - \frac{\sum_{m=1}^{N_p} \psi_{kijm}\mathcal{M}(\mathbf{x}_{k-1}^{(m)})}{\sum_{m=1}^{N_p} \psi_{kijm}}\right], \tag{11}$$

If the model is stochastic with additive Gaussian errors, eq. (11) is exact (Pulido and van Leeuwen, 2019).

To illustrate the practical implementation of this approach, we evaluate eq. (5) under the assumption of a Gaussian prior distribution with a radial basis function kernel,

$$\mathbf{v}(\mathbf{x}) = \frac{1}{N_p}\sum_{j=1}^{N_p}\left\{e^{-\frac{1}{2}\left\|\mathbf{x}_{k,i-1}^{(j)} - \mathbf{x}\right\|_\Sigma^2}\left[\mathbf{H}^\top \mathbf{R}_k^{-1}\left(\mathbf{y}_k - \mathcal{H}\left(\mathbf{x}_{k,i-1}^{(j)}\right)\right) - \mathbf{B}_k^{-1}\left(\mathbf{x}_{k,i-1}^{(j)} - \overline{\mathbf{x}}_{k,0}\right) - \Sigma_k^{-1}\left(\mathbf{x}_{k,i-1}^{(j)} - \mathbf{x}\right)\right]\right\}. \tag{12}$$

The computational cost of a single pseudo-time iteration in the Gaussian-mixture prior case is

$$\mathcal{O}\left(N_x^2 N_p^2 + N_y^2 N_p + N_x N_y N_p\right), \tag{13}$$

where the first term corresponds to the kernel and its gradient calculation, while the second and third terms correspond to the computational cost of the likelihood. Assuming that the matrix inversion can be performed in $\mathcal{O}_{\text{inv}}(N_x)$, the overall computational cost becomes

$$\mathcal{O}\big(\mathcal{O}_{\text{inv}}(N_x) + N_{it}\left[N_x^2 N_p^2 + N_y^2 N_p + N_x N_y N_p\right]\big),\tag{14}$$

where $N_{it}$ denotes the number of pseudo-time iterations.

## 2.2 Localization methods

The underlying assumption in the two developed localization methods is that on average, error correlations decay with the physical distance so that when the distance between two variables is larger than a given threshold known as the localization radius, the correlation is assumed to be negligible. The correlations of these far points are neglected so that it becomes feasible to produce an inference using only the points of the background state and the observations within the localization radius. This reduction in algorithmic complexity allows to reduce sampling noise and enhance the quality of the analysis for high-dimensional state spaces.

Both Hu and van Leeuwen (2021) and the present work address the challenge of applying MPF in high-dimensional systems, but through different approaches. Hu and van Leeuwen focus on an intrinsic modification of the particle interaction mechanism by transitioning from a scalar kernel to a matrix kernel; within the matrix kernel, they assume the distance between particles is independent for each component of the state vector. In addition to the kernel modification, their work also applies localization to the prior covariance matrix through a Schur product with a distance-decaying correlation matrix. In contrast, this work starts from a localization assumption, and applies it coherently to both the posterior distribution and the kernel. This results in explicit localization schemes that restructure how the optimization process is applied in state space, thereby modifying the sequencing of the optimization process.

The $\alpha$-localization algorithm assumes that the kernels are localized around each variable, so that distances between particles are measured in a low dimensional space. Furthermore, it uses a localized prior covariance matrix, for instance, by keeping blocks of the global sample covariance via the Schur product. The state updates are determined globally.

On the other hand, the $\beta$-localization algorithm assumes the full local variational mapping process is localized around each variable. In terms of the localization assumption, this method is similar to the localization in Hunt et al. (2007) and has also some resemblance to the methodology implemented in Hu et al (2024). For each variable, the optimization is conducted separately considering the observations and the prior state variables within the localization radius.

### 2.2.1 $\alpha$-Localization

Given a variable $x_l$ of the state $\mathbf{x}$, we consider a neighborhood $\mathcal{C}_l$ of $x_l$ and denote the variables within this neighborhood as $\tilde{\mathbf{x}}_l = \{x_{l'}; l' \in \mathcal{C}_l\}$. We assume that the variables located outside of $\mathcal{C}_l$ are statistically independent of $x_l$,

$$p(x_l|\mathbf{x}) = p(x_l|\tilde{\mathbf{x}}_l).\tag{15}$$

For simplicity, we assume a single physical type of variable in the state space. In this approach, the local state $\tilde{x}_{k,i,l}^{(j)}$ is defined with four indices: $k$ (time index), $i$ (pseudo-time iteration index), $l$ (space index) and $j$ (particle index). To avoid overclutter, time and iteration indices are omitted. In a one dimensional space for a localization radius $\ell$, the vector of neighbor variables is $\tilde{\mathbf{x}}_l = \{x_{l'}; l - \ell \le l' \le l + \ell\}$ with dimension $N_{\tilde{\mathbf{x}}} = 2\ell + 1$.

The global update of variable $x_l$ from eq. (5) is

$$v_l(\mathbf{x}) = \frac{1}{N_p} \sum_{j=1}^{N_p} \left[ K(\mathbf{x}^{(j)}, \mathbf{x}) \left( \partial_{x_l^{(j)}} \log p(\mathbf{x}^{(j)}) + \partial_{x_l^{(j)}} \log K(\mathbf{x}^{(j)}, \mathbf{x}) \right) \right]. \tag{16}$$

where $\partial_{x_l^{(j)}}$ is the partial derivative with respect to the $l$ variable. Since the variables beyond the localization radius are assumed to be statistically independent, we approximate eq. (16) by considering a local kernel following Wang et al. (2018) in which only the variables within the localization radius around $l$ are considered. This local kernel is denoted as $K_l(\tilde{\mathbf{x}}_l^{(j)}, \tilde{\mathbf{x}}_l)$.

The local kernel is specific to each grid point $x_l$. It calculates the Mahalanobis distance between particles using a state vector that is centered at $x_l$ and includes only the neighboring points that fall within the defined localization radius.

The local kernel is defined with a radial basis function as the global one, but with a kernel covariance matrix defined as the Schur (element-wise) product of a localization matrix and the global covariance matrix, $\Lambda_l = \Gamma_l \circ \Sigma$, where the localization matrix $\Gamma_l$ could be a block matrix around $l$ with one's and zeros, as in eq. (17), or some decaying coefficient with the distance of the rest of the points to the $l$-th grid point (e.g. Gaspari and Cohn (1999) factor). The neighborhood variables $\tilde{\mathbf{x}}_l$ are the ones where $\Gamma_l$ is not null.

$$(\Gamma_l)_{mn} = \begin{cases} 1 & \text{if} \quad l - \ell \le m, n \le l + \ell \\ 0 & \text{otherwise} \end{cases} \tag{17}$$

The resulting local kernel is:

$$K_l(\tilde{\mathbf{x}}_l, \tilde{\mathbf{x}}_l') = e^{-\frac{1}{2} \left\| \tilde{\mathbf{x}}_l - \tilde{\mathbf{x}}_l' \right\|_{\Lambda_l}^2} \tag{18}$$

The crucial feature of the local kernel is that the Mahalanobis distance calculation only takes into account low-dimensional states. The local flow in the $l$ variable is therefore approximated by

$$v_l(\tilde{\mathbf{x}}_l) \approx \frac{1}{N_p} \sum_{j=1}^{N_p} \left[ K_l(\tilde{\mathbf{x}}_l^{(j)}, \tilde{\mathbf{x}}_l) \left( \partial_{x_l^{(j)}} \log p(\tilde{\mathbf{x}}_l^{(j)}) + \partial_{x_l^{(j)}} \log K_l(\tilde{\mathbf{x}}_l^{(j)}, \tilde{\mathbf{x}}_l) \right) \right] \tag{19}$$

The gradient of the posterior density will be calculated following eq. (7) with the following modifications. For the gradient of the likelihood term, it is calculated globally using the first term of eq. (7), resulting in a matrix in $\mathbb{R}^{N_x \times N_p}$. The term used in eq. (19) corresponds to the $l$-th row of that global matrix. For the prior density term, we calculate it according to eq. (9) or eq. (11) (depending on our hypothesis), but we use the localized vector $\tilde{\mathbf{x}}_l$ and apply the localized covariance matrix $\Gamma_l \circ \mathbf{B}_k$ or $\Gamma_l \circ \mathbf{Q}_k$. This approach is the same as in the local kernel calculation in eq. (18).

We could also derive the local velocity, eq. (19), under the assumption that state-space covariance matrices are l-block diagonal while keeping the state vectors global. In preliminary experiments, we evaluated a hybrid methodology in which

the kernel was local, with a block $\Lambda_l$ matrix, but the prior density was global with l-banded background covariance matrices. However the performance of this hybrid methodology was suboptimal.

Once the complete velocity vector is reconstructed with each component computed separately, the global states in the next pseudo-time are determined by eq. (4). Therefore, covariance inversion and the mappings are global. In algorithm 1 below, a pseudocode of the $\alpha$-localization algorithm is presented. In this case, the time index is omitted, while the pseudo-time, space, and particle indices are retained. The computational complexity of the LMPF-$\alpha$ using a Gaussian-mixture prior is $\mathcal{O}\big(\mathcal{O}_{\text{inv}}(N_x,\ell) + N_{it}\left[N_x(2\ell+1)^2 N_p^2 + N_y^2 N_p + N_x N_y N_p\right]\big)$, where $\mathcal{O}_{\text{inv}}(N_x,\ell)$ is the cost of inverting an $\ell$-banded matrix.

### 2.2.2 $\beta$-Localization

The $\beta$-localization involves a physical partitioning of the state space centred around each variable based on distance between variables. Subsequently, it leverages the same principles of the global MPF to each partition.

This methodology is based on Zhuo et al. (2018) in which the KL divergence is decomposed as,

$$D_{KL}(q||p) = D_{KL}\left(q(x_l|\mathbf{x}_{\neg l})q(\mathbf{x}_{\neg l})||p(x_l|\tilde{\mathbf{x}}_l)p(\mathbf{x}_{\neg l})\right) + D_{KL}\left(q(\mathbf{x}_{\neg l})||p(\mathbf{x}_{\neg l})\right) \tag{20}$$

where $\mathbf{x}_{\neg l}$ is composed by all the state variables except $x_l$. Therefore, we can solve a local minimization problem for $x_l$ to find $q(x_l|\tilde{\mathbf{x}}_l)$ and by keeping fixed the rest, $q(\mathbf{x}_{\neg l})$.

This approach guarantees that the analysis is performed independently for each state variable, with no dependency on intermediate updates of other grid points. However, the neighborhood variables are considered to define the map for each state variable. This means that while the local analysis at a given grid point depends on nearby observations, the convergence at each point remains independent. This reminds the application of normalizing flows with transformations in each direction (Tabak and Turner, 2012). These local minimizations are iterated along $l$.

The $\beta$-localization algorithm consists of applying the global MPF to the neighborhood of $x_l$. For a given localization radius $\ell$, we use the neighborhood vector as in the $\alpha$-localization: $\tilde{\mathbf{x}}_l$. The local velocity is defined as the global velocity in eq. (5), but calculated only over the localized state vector $\tilde{\mathbf{x}}_l$. Therefore, it considers a kernel as in eq. (6) calculated in the physically partitioned state. A localized posterior density is also used, in which only the forecast states in the local domain $\tilde{\mathbf{x}}_l^{f(j)} \in N_{\tilde{\mathbf{x}}_l}$ are considered. Observations within the localization radius are selected. This localization algorithm can only be applied for observations that have a well-defined location in physical space. For that purpose we define $\mathcal{I}_l$ as the set of observation indices corresponding to the observations that are relevant to the localized state vector $\tilde{\mathbf{x}}_l$. Specifically, for a one-dimensional domain this is

$$\mathcal{I}_l = \{m \mid \text{the position of observation } y_m \text{ lies within the interval } [l - \ell, l + \ell]\}. \tag{21}$$

The localized observation vector $\tilde{\mathbf{y}}_l$ is then defined as the subset of observations whose indices belong to $\mathcal{I}_l$:

$$\tilde{\mathbf{y}}_l = \{y_m \mid m \in \mathcal{I}_l\}. \tag{22}$$

The blocks of $\tilde{\mathcal{H}}_l \in N_{\tilde{\mathbf{y}}_1} \times N_{\tilde{\mathbf{x}}_1}$ and $\mathbf{R}_l \in N_{\tilde{\mathbf{y}}_1} \times N_{\tilde{\mathbf{y}}_1}$ are also coherently selected,

$$\tilde{\mathcal{H}}_l = (\tilde{\mathcal{H}})_{mn} \text{ with } m \in \mathcal{I}_l, n = l - \ell, \dots, l + \ell, \tag{23}$$

$$\mathbf{R}_l = (\mathbf{R})_{mn} \text{ with } m, n \in \mathcal{I}_l. \tag{24}$$

Thus, we are using observations from a subspace and their associated error covariance related to that subset of observations.
Additionally, for large localization radii where distant spurious covariances might still occur, a length-decaying factor could be useful.

For each grid point in the domain, i.e. state variable, the following iterative transformation is applied:

$$\tilde{\mathbf{x}}_{i,l}^{(j)} = \tilde{\mathbf{x}}_{i-1,l}^{(j)} + \epsilon \mathbf{v}_l(\tilde{\mathbf{x}}_{i-1,l}^{(j)}) \tag{25}$$

The convergence is independent for each grid point. To obtain the global analysis vector, only the element at position $l$ from this local analysis vector is kept. This process is repeated for every spatial point on the grid, i.e. variable of the state vector. When updating a given grid point in the $\beta$ approach, the values of all other grid points are taken from the original prior state of the particle, not from previously updated points in the same cycle. This avoids any dependence on the order in which the domain is processed.

The order of pseudo-time iterations and localized step iterations is reversed between the two methodologies. In the $\alpha$ approach, for each pseudo-time step, the entire domain is updated, resulting in a global state for each pseudo-time step. In contrast, in the $\beta$ approach, for each local point in the domain, all pseudo-time steps are iterated independently before moving to the next state variable, leading to localized convergence without a global state. The exchange of iterations is easier to observe by looking at the algorithms of LMPF-$\alpha$ in algorithm 1 and LMPF-$\beta$ in algorithm 2.

---

**Algorithm 1** LMPF-$\boldsymbol{\alpha}$: Global update

Compute global $\Sigma$
\# Number of pseudo time step iterations is denoted as $N_{it}$
**for** $i = 1$ to $N_{it}$ **do**
  **for** $l = 1$ to $N_x$ **do**
    $\tilde{\mathbf{x}}_{i-1,l}^{(j)} \leftarrow x_{i-1,m}^{(j)}$, with $m = l - \ell, \dots, l + \ell$ and $j = 1, \dots, N_p$
    Compute $\Lambda_l^{-1}$
    Compute localized log posterior
    Compute $v_l^{(j)}\left(\tilde{\mathbf{x}}_{i-1,l}^{(1:N_p)}\right)$ as in eq. (19)
    $x_{i,l}^{(j)} \leftarrow x_{i-1,l}^{(j)} + \epsilon v_l^{(j)}\left(\tilde{\mathbf{x}}_{i-1,l}^{(1:N_p)}\right)$
  **end for**
  \# Global state updated at pseudo-time step $i$
**end for**

---

**Algorithm 2** LMPF-$\boldsymbol{\beta}$: Local update

**for** $l = 1$ to $N_x$ **do**
  $\tilde{\mathbf{x}}_{0,l}^{(j)} \leftarrow x_{0,m}^{(j)}$, with $m = l - \ell, \dots, l + \ell$ and $j = 1, \dots, N_p$
  **for** $i = 1$ to $N_{it}$ **do**
    Compute $\Sigma^{-1}$ in the local set $\{\tilde{\mathbf{x}}_{i-1,l}^{(1:N_p)}\}$
    Compute $\mathbf{v}_l^{(j)}\left(\tilde{\mathbf{x}}_{i-1,l}^{(1:N_p)}\right)$
    $\tilde{\mathbf{x}}_{i,l}^{(j)} \leftarrow \tilde{\mathbf{x}}_{i-1,l}^{(j)} + \epsilon \mathbf{v}_l^{(j)}\left(\tilde{\mathbf{x}}_{i-1,l}^{(1:N_p)}\right)$
  **end for**
  Retain center value from $\tilde{\mathbf{x}}_{N_{it},l}^{(j)}$
  \# Local convergence at point $l$
**end for**

The computational cost of the LMPF-$\beta$ is $\mathcal{O}\big(N_x \mathcal{O}_{\text{inv}}(2\ell+1) + N_x N_{it}\left[(2\ell+1)^2 N_p^2 + N_y'^2 N_p + (2\ell+1)N_y' N_p\right]\big)$, where $N_y'$ denotes the maximum number of observations within any localized domain. This is equivalent to $\mathcal{O}(N_x)\mathcal{O}_{\text{MPF}}(2\ell+1, N_y', N_p)$, which represents the computational complexity of the global MPF algorithm.

## 3 Numerical setup

The global MPF and the two variants of the localized MPF are assessed in experiments with synthetic observations. In these experiments, observations are generated based on a known dynamical model. The true state is the solution of the known model, referred to here as the nature model. In contrast, the forecast model is a surrogate for the nature model, so we consider the assimilation experiments in the presence of model error. The surrogate model is used to produce forecasts within the assimilation system, $\mathbf{x}_k^{f(j)} = \mathcal{M}^{su}(\mathbf{x}_{k-1}^{a(j)})$. After evolving the previous analysis ensemble states, $\mathbf{x}_{k-1}^{a(j)}$ with this surrogate model, the data assimilation step is conducted, and so on. This approach allows us to examine the assimilation scheme with a known true state in the presence of model errors. In these proof-of-concept experiments, the nature model is the two-scale Lorenz system (section 3.1), while the surrogate model is the one-scale Lorenz (Section section 3.2) so that the source of model errors is the lack of the explicit representation of small-scale dynamics. Both models are deterministic, with no explicit additive stochastic error terms.

### 3.1 Description of the true model

The nature model is defined by the two-scale Lorenz system equations (Lorenz, 2005):

$$\begin{cases} \dot{X}_n = -X_{n-1}(X_{n-2} - X_{n+1}) - X_n + F - \frac{hc}{b}\sum_{j=J(n-1)+1}^{nJ} Y_j & n = 1,\cdots,N_{LS} \\ \dot{Y}_m = -cbY_{m+1}(Y_{m+2} - Y_{m-1}) - cY_m + \frac{hc}{b}X_{\text{int}[(m-1)/J]+1} & m = 1,\cdots,N_{SS} \end{cases} \tag{26}$$

within a cyclic domain, i.e, $X_{N_{LS}+1} = X_1, X_0 = X_{N_{LS}}$, and $X_{-1} = X_{N_{LS}-1}; Y_{N_{SS}+1} = Y_1, Y_{N_{SS}+2} = Y_2$ and $Y_0 = Y_{N_{SS}}$. $N_{LS}$ is the number of large-scale ($LS$) variables, and $N_{SS}$ the number of the small-scale ($SS$) variables. The equations are solved using a fourth-order Runge-Kutta scheme. The parameters of the nature model are specified in table 1. They correspond to the standard configuration of the two-scale Lorenz system following Wilks (2005).

### 3.2 Description of the surrogate model

The forecast model employed in the data assimilation system is the corresponding one-scale Lorenz system (Lorenz and Emanuel, 1998). This model exclusively replicates the large-scale equations so that the influence of the small-scale variables must be parameterized. As the true model, the equations are solved using a fourth-order Runge-Kutta scheme. The equations for the one-scale case are

$$\dot{X}_n = -X_{n-1}(X_{n-2} - X_{n+1}) - X_n + f(X_n) \quad n = 1,\cdots,N_{SU} \tag{27}$$

within a cyclic domain. $N_{SU}$ is the number of variables of the surrogate model. In order to be consistent, $N_{SU}$ must be equal to $N_{LS}$. The external forcing, $f(X_n)$, is defined as $f(X_n) = F + f^*(X_n)$ and consists of a linear parameterization of the effects of

small-scale dynamics. The parameterization coefficients, $F$ and $f^*$, are estimated using the methodology proposed by Pulido et al. (2016). The parameters of the forecast model are specified in table 2.

<table>
<tr><td colspan="3"><b>Table 1.</b> True model parameters.</td><td colspan="3"><b>Table 2.</b> Surrogate model parameters.</td></tr>
<tr><td>Variable</td><td>Value</td><td>Variable Name</td><td>Variable</td><td>Value</td><td>Variable Name</td></tr>
<tr><td>$N_{LS}$</td><td>40</td><td>Large-scale dimension</td><td>$N_{SU}$</td><td>40</td><td>Surrogate model dimension</td></tr>
<tr><td>$N_{SS}$</td><td>1280</td><td>Small-scale dimension</td><td>$f(X_n)$</td><td>$26 + f^*(X_n)$</td><td>Forcing terms</td></tr>
<tr><td>J</td><td>32</td><td>$N_{LS}/N_{SS}$</td><td>$f^*(X_n)$</td><td>$0.73 \cdot X_n + 0.91$</td><td>Parameterized forcing</td></tr>
<tr><td>F</td><td>26</td><td>External forcing</td><td>$dt$</td><td>$5 \times 10^{-3}$</td><td>Time integration step</td></tr>
<tr><td>c</td><td>10</td><td>Time scale-ratio</td><td></td><td></td><td></td></tr>
<tr><td>b</td><td>10</td><td>Space scale-ratio</td><td></td><td></td><td></td></tr>
<tr><td>h</td><td>1</td><td>Coupling constant</td><td></td><td></td><td></td></tr>
<tr><td>dt</td><td>$1.25 \times 10^{-3}$</td><td>Time integration step</td><td></td><td></td><td></td></tr>
</table>

## 3.3 Experimental setup

### 3.3.1 Initial state and observations

To generate the synthetic observations, an initial true state $\mathbf{x}_0^t$ is obtained after integrating the nature model from a random initial condition over a long period. The nature model is then evolved from this initial true state for $N_t = 10,000$ cycle times, where one cycle corresponds to the observation interval of 0.05 time units and consists of 40 integrations of the nature model. Observations are then generated from the large-scale part ($LS$) of the true states $\left(\mathbf{x}_k^t\big|_{LS}\right)$ at each cycle,

$$\mathbf{y}_k = \mathcal{H}\left(\mathbf{x}_k^t\big|_{LS}\right) + \boldsymbol{\nu}_k, \tag{28}$$

where observational errors are unbiased with variance $\mathbf{R}_k$, i.e. $\boldsymbol{\nu}_k \sim \mathcal{N}(\mathbf{0}, \mathbf{R}_k)$ and $\mathbf{x}_k^t$ represents the evolution of the nature model $\mathbf{x}_k^t = \mathcal{M}^t(\mathbf{x}_{k-1}^t)$. The observation operator is assumed to be constant over time. We assume that the observational covariance matrix is also fixed, and diagonal, i.e.

$$\mathbf{R}_k = \sigma_R^2 \cdot \mathbf{I}_{N_y \times N_y} \tag{29}$$

Three different observational operators are used: A linear operator $\mathcal{H}$, where $\mathcal{H}(x) = x$ and $\sigma_R^2 = 0.5$. A square operator $\mathcal{H}$, where $\mathcal{H}(x) = x^2$ and $\sigma_R^2 = 0.5$. A logarithmic operator $\mathcal{H}$, where $\mathcal{H}(x) = \log(|x|+1)$ and $\sigma_R^2 = 0.05$. The logarithmic operator $\log(|x|+1)$ was chosen instead of $\log(|x|)$ because, for values of $x$ close to zero, the observation operator may diverge and worsen the performance of the assimilation, making it necessary to apply a quality control routine. Also, following Kurosawa and Poterjoy (2021), a smaller observation error is used to avoid filter divergence.

Experiments for each observation operator were conducted with full observations (that is, $N_y = 40$) and with partial observations (that is, $N_y = 20$ with observations at every other grid point). In addition, each combination of observation operator and observation network was run with $N_p = 20$ and $N_p = 50$ particles.

To set the $N_p$ initial states of the particles, we use randomly chosen times from a long simulation of the surrogate model. This selection is used to create the first ensemble, whose particles are independent of the initial true state.

### 3.3.2 Specifications of the MPF

As mentioned, a Gaussian radial basis function is used as the kernel in eq. (6), with its covariance matrix taken to be proportional to the forecast covariance estimated from the sample,

$$\Sigma = \gamma \cdot \hat{\mathbf{P}}^f = \frac{\gamma}{N_p - 1} \sum_{j=1}^{N_p} (\mathbf{x}^{f(j)} - \overline{\mathbf{x}}^f)(\mathbf{x}^{f(j)} - \overline{\mathbf{x}}^f)^\top \tag{30}$$

where $\gamma$ is a bandwidth hyperparameter and $\overline{\mathbf{x}}^f$ denotes the sample mean of the forecasts across the particles. In this work, we tune this $\gamma$ hyperparameter for the experiments using a brute-force search. The step size of the mapping $\epsilon$ is determined adaptively using the Adam optimization method (Kingma and Ba, 2014) with up to 500 iterations of pseudo-time in each cycle.

For two key experimental setups—the fully observed linear case and the partially observed logarithmic case—we first performed sensitivity analysis by varying the localization radius, which led us to establish a default value of $\ell = 3$ for all subsequent experiments. We then conducted additional sensitivity tests for these same two scenarios to assess performance dependence on particle number. Finally, for these two scenarios, to evaluate algorithm behavior under large model error conditions, we conducted experiments where the linear parameterization was omitted from the surrogate model, significantly increasing the model error.

A non-Gaussian posterior density may be the result of a non-linear observation operator or a non-Gaussian prior density distribution resulting from non-linear forecasts. One of the objectives of this work is to evaluate the performance of the MPF in experiments with two prior density distributions: a Gaussian and a Gaussian mixture. In the Gaussian experiments, the resulting gradient of the logarithm of the prior density function is given by eq. (9), in which we take $\mathbf{B} = \hat{\mathbf{P}}^f$. In the global and $\alpha$-localization cases, this matrix is scaled by a Gaspari-Cohn decaying factor. However, in $\beta$-localization, scaling of the prior covariance matrix is not required for small localization radii and thus will not be applied.

In the Gaussian mixture experiments, we use the expression given in eq. (11) for the density. The matrix $\mathbf{Q}_k$ is defined as $\mathbf{Q}_k = \xi \cdot \mathbf{P}^f$ where $\xi$ is a bandwidth hyperparameter of the mixtures. Tuning this hyperparameter contributes to enhancing the performance of the MPF. The number of Gaussians corresponds to the number of particles.

In preliminary experiments, we found that a multiplicative or additive inflation factor is not required in the MPF even when applied over an extended period. In fact, adding an inflation factor degraded the performance of the filter.

## 4 Results

In each experiment, a comparison is made between the global MPF, both localization schemes, the Ensemble Transform Kalman Filter (ETKF, Bishop et al., 2001) and the LETKF with $\ell = 3$. We compare the LMPFs against the LETKF and ETKF as these represent classical, computationally efficient ensemble filters that provide good baseline performance and are widely used in operational data assimilation systems.

The Root Mean Square Error (RMSE) between the true state and the analysis ensemble mean, and the spread of the analysis ensemble are the primary metrics used to compare the performance of each experiment. The time series consists of 10,000 cycles, with the initial 1,000 cycles designated as the spin-up period and excluded from the analysis. The temporal averages of the RMSE and spread are then calculated over the subsequent 9,000 cycles. Besides, to evaluate the dependence on the initial conditions, for each experiment, 10 realizations were conducted. The results show the mean of these realizations and the error bar represent the sample standard deviation.

Before the experiments, we conducted a hyperparameter optimization. In the case of Kalman filters, this involves a multiplicative inflation factor that minimizes the RMSE of the analyses.

For the MPF and its local versions, one of the key hyperparameters is the proportionality factor of the kernel sample covariance $\gamma$. For experiments assuming a Gaussian mixture, another hyperparameter is the width of the Gaussian mixtures, $\xi$. Thus, the optimization is performed in the 2D space defined by $\gamma$ and $\xi$ for the Gaussian mixture case by brute force. The hyperparameters are selected to minimize RMSE in order to examine how the methodologies represent ensemble spread under optimal RMSE conditions. The goal is to evaluate whether the methodologies produce a reasonable spread representation at their lowest RMSE without explicitly tuning for it. As an example of the hyperparameter tuning, fig. 1 shows the optimization of the global MPF with Gaussian mixture prior in a fully observed linear case using 20 particles. As illustrated in fig. 1, the dependence of RMSE on $\gamma$ and $\xi$ is nontrivial and non-intuitive, which prevents the definition of a simple rule of thumb. Optimization for the different variants of the MPF and the ensemble Kalman filters is performed for each particle size and observation network. In Appendix A we present the optimal parameters for some of the experiments.

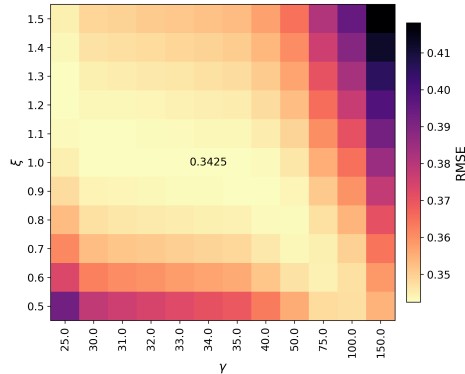

**Figure 1.** Time and variable averaged RMSE for the MPF experiment as a function of the bandwidth of the Gaussian mixtures $\xi$ and the bandwidth of the kernel $\gamma$.

For the experiments shown in this work with the two-scale Lorenz system and its surrogate one-scale Lorenz model, a model integration without data assimilation achieves an RMSE of 6.78 and a spread of 6.55. The RMSE of 6.78 represents the maximum error of the forecast model without assimilation, providing a top value for evaluating the impact of incorporating observational data in the assimilation process. Hereafter, we refer to this value as the NoDA-RMSE.

## 4.1 Linear observational operator

The first experiment evaluates the performance of the local mapping particle flow filters under a linear observation operator, as in eq. (28). Figure 2 shows the results for the fully observed (left panels) and partially observed (right panels) scenarios, employing 20 in fig. 2a and 50 particles in fig. 2b. Black dots and crosses represent Gaussian filters or MPFs that assume Gaussian priors. Red dots and crosses represent particle flow filters with Gaussian mixture priors.

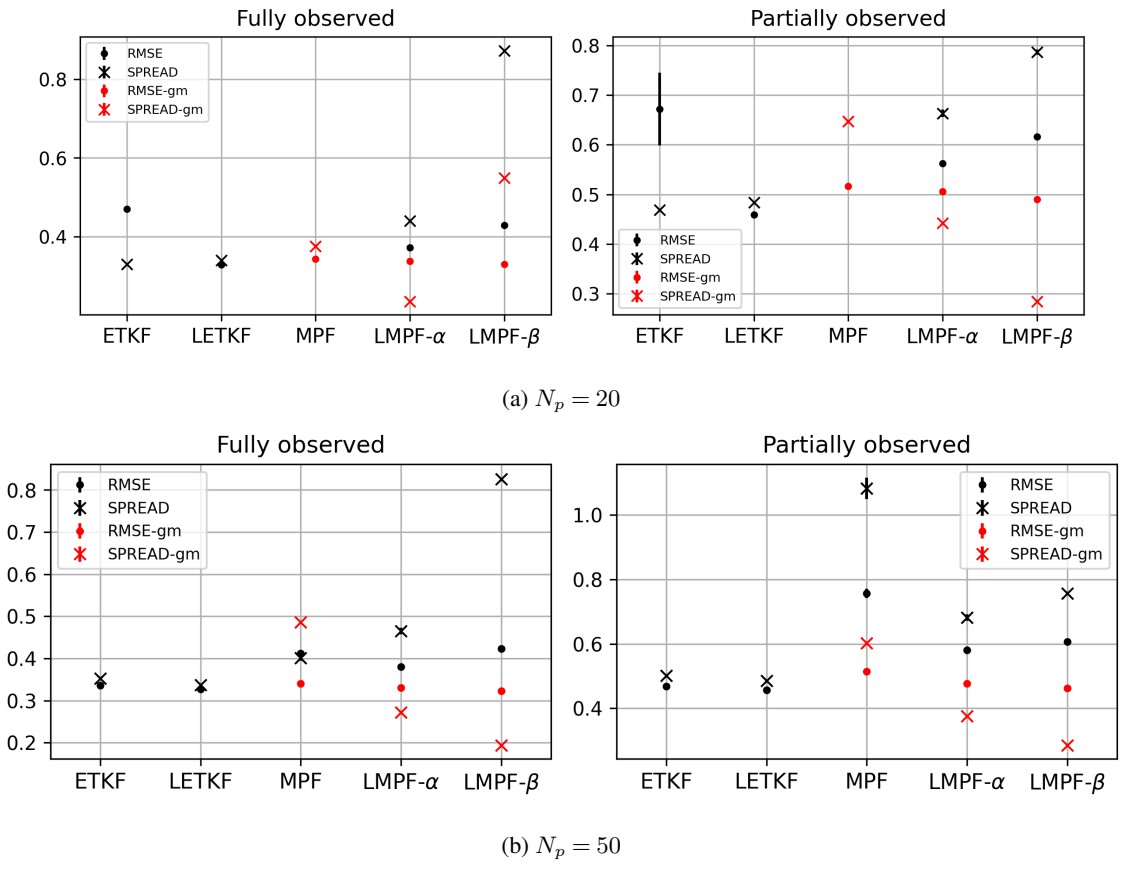

**Figure 2.** RMSE and spread in the linear observation operator for 20 and 50 particles, under both fully and partially observed scenarios.

All MPF experiments exhibit better performance than ETKF for the 20-particle experiments with a full observed state, as in fig. 2a, except for the global MPF using a pure Gaussian prior PDF. This last case converges to an RMSE of $0.644\pm0.001$

but with an extremely high spread value (6.70±0.04). On the other hand, when a Gaussian mixture prior density is utilized, represented by the red dots, all three MPF experiments demonstrate performances similar to LETKF.

Regarding the spread, MPF and LMPF with pure Gaussian priors tend to have large dispersion. In the case of LMPF-$\beta$, the spread is 0.92 and is not shown. However, this pattern changes in Gaussian mixture experiments, where the spread is much closer to the RMSE. The global case provides the spread that is closest to the RMSE. It is important to note that these spread results come from experiments using optimal hyperparameters in terms of RMSE.

Despite the linearity of the observational operator, the model dynamics is non-linear. Consequently, it is expected that Gaussian mixtures capture non-linearities more effectively compared to experiments utilizing pure Gaussian priors. This could explain the better performance of the Gaussian mixture experiments.

The right panel of fig. 2a shows results for partially observed experiments. In the Gaussian prior case, the global MPF converged to a very high RMSE (1.85±0.03) and is therefore not shown. The localized filters achieve a lower RMSE than ETKF, and perform similarly than LETKF.

As in the fully observed scenario, the Gaussian mixture experiments show a significant improvement across all MPFs. The resulting RMSE is comparable to that of LETKF.

Figure 2b presents the results for the 50-particle experiments. The performance relationships among the experiments are similar to the previous case, with the notable exception of ETKF, which shows the most significant improvement. In this case, the Gaussian-prior global MPF achieves convergence, although its RMSE remains higher than that of the Kalman filters. Similarly, the Gaussian mixture experiments demonstrate a significant improvement in RMSE.

In the partially observed scenario, the ETKF demonstrates the most significant improvement, and the Gaussian-prior MPF successfully converges. Additionally, the spread of the Gaussian-mixture prior MPF is closer to the RMSE in the global case. The localized particle filters exhibit a similar behavior to that observed in the fully observed case.

We note that these experiments use a localization radius of $\ell = 3$ which is only optimal for the LETKF with $N_p = 20$. The localization radius of $\ell = 3$ was fixed in all the experiments to ensure fair comparison across all methods and ensemble sizes, recognizing that this choice may not be individually optimal for each configuration. Notably, the LMPFs exhibit better performance for longer radii, so that the $\ell = 3$ fixed radius choice does not systematically favor the proposed localization methods. This setup ensures that differences in performance can be attributed solely to the algorithms themselves rather than variations in the localization radius.

## 4.2 Square observational operator

A square observational operator presents a challenge for data assimilation schemes, as it treats negative and positive true states with the same absolute value as equivalent, so that the error distributions in the hidden state space are likely to be a bimodal distribution.

Figure 3a presents the square-$\mathcal{H}$ results for the 20-particle experiments. Overall, the RMSE values are smaller compared to the linear case. This difference is linked to the choice of model error variance. While $\sigma_R^2 = 0.5$ in both cases, the magnitude

of nonlinear observations is typically much greater than that of linear observations, resulting in a relatively smaller error in the nonlinear case.

In the fully observed case, both the ETKF and the global MPF with a Gaussian prior converged to very high RMSE values. The Gaussian-prior MPF achieved an RMSE>NoDA-RMSE. However, the LETKF and Gaussian-prior LMPFs achieve good RMSE performance, though with a significant underestimation of the spread in the LMPF-$\alpha$. On the other hand, the three experiments employing Gaussian-mixture priors demonstrate very good performance, similar to the LETKF. The impact of localization is pronounced in the ensemble Kalman filters (as seen in the ETKF vs. LETKF performance) but has only a minor effect on the Gaussian-mixture MPFs.

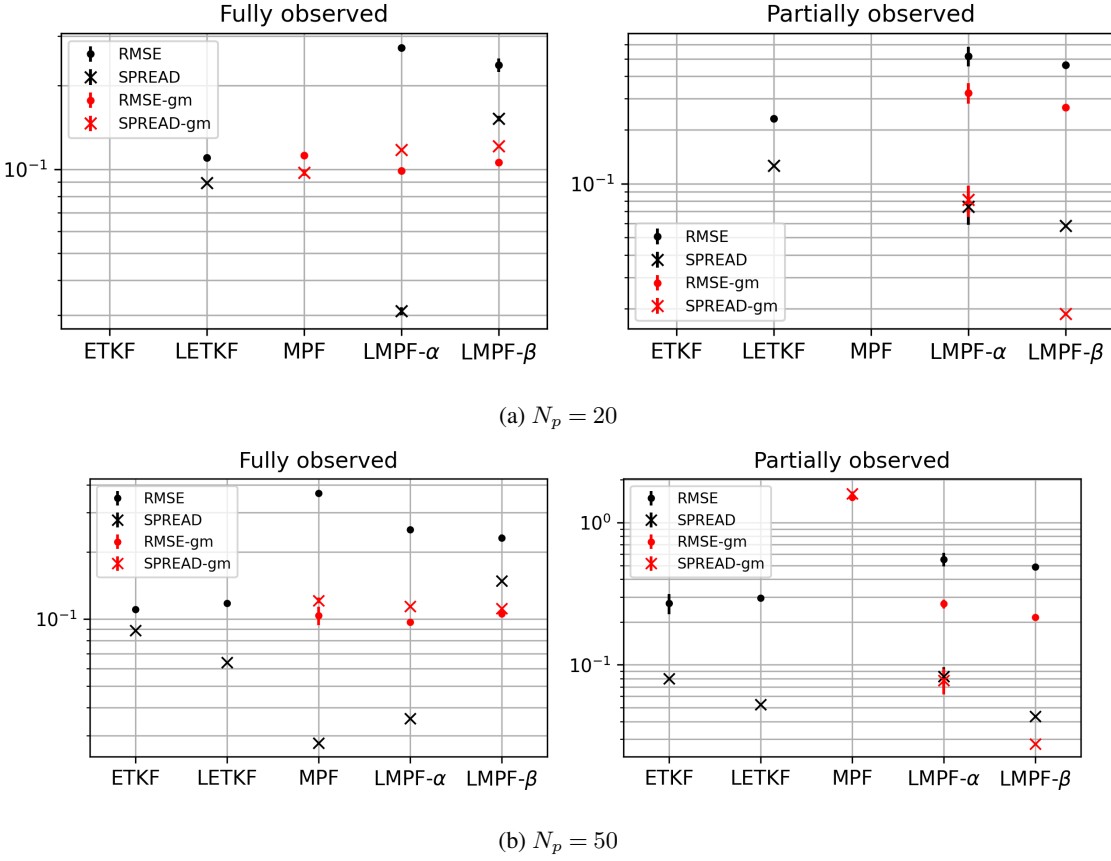

(a) $N_p = 20$

(b) $N_p = 50$

**Figure 3.** RMSE and spread for the experiments with a square observation operator for 20 (a) and 50 (b) particles, under both fully (left panels) and partially observed (right panels) scenarios.

In the partially observed case, the ETKF diverged for all inflation parameters tested and the Gaussian-prior MPF achieved an RMSE>NoDA-RMSE. LETKF shows a similar RMSE than the Gaussian-mixture particle filters. Figure 3b presents the results for the 50-particle experiments. In the fully observed scenario, the Gaussian-prior MPF successfully converges, unlike in the 20-particle case, but with a high RMSE for this observational operator (0.367±0.007) and a significant low spread.

The ETKF demonstrates a notable improvement in accuracy, outperforming its localized version. A similar effect is observed in the Gaussian-mixture experiments, where the MPF achieves similar accuracy than its localized counterparts. In this case, the Gaussian-mixture particle filters provide a higher spread compared to the Gaussian-prior filters, with the exception of the Gaussian-prior LMPF-$\beta$.

In the partially observed scenario, the ETKF achieves convergence with an RMSE similar to that of LETKF. Once again, the Gaussian-mixture filters demonstrate the best performance, comparable to the Kalman filters, with the exception of the global MPF, which showed a very high RMSE value. In this case, the effect of localization is very positive.However, as in the 20-particle case, all filters significantly underestimate the spread.

## 4.3 Logarithmic observational operator

Figure 4 shows the performance of the filters in the logarithmic observation operator case, assessing a highly non-Gaussian regime.

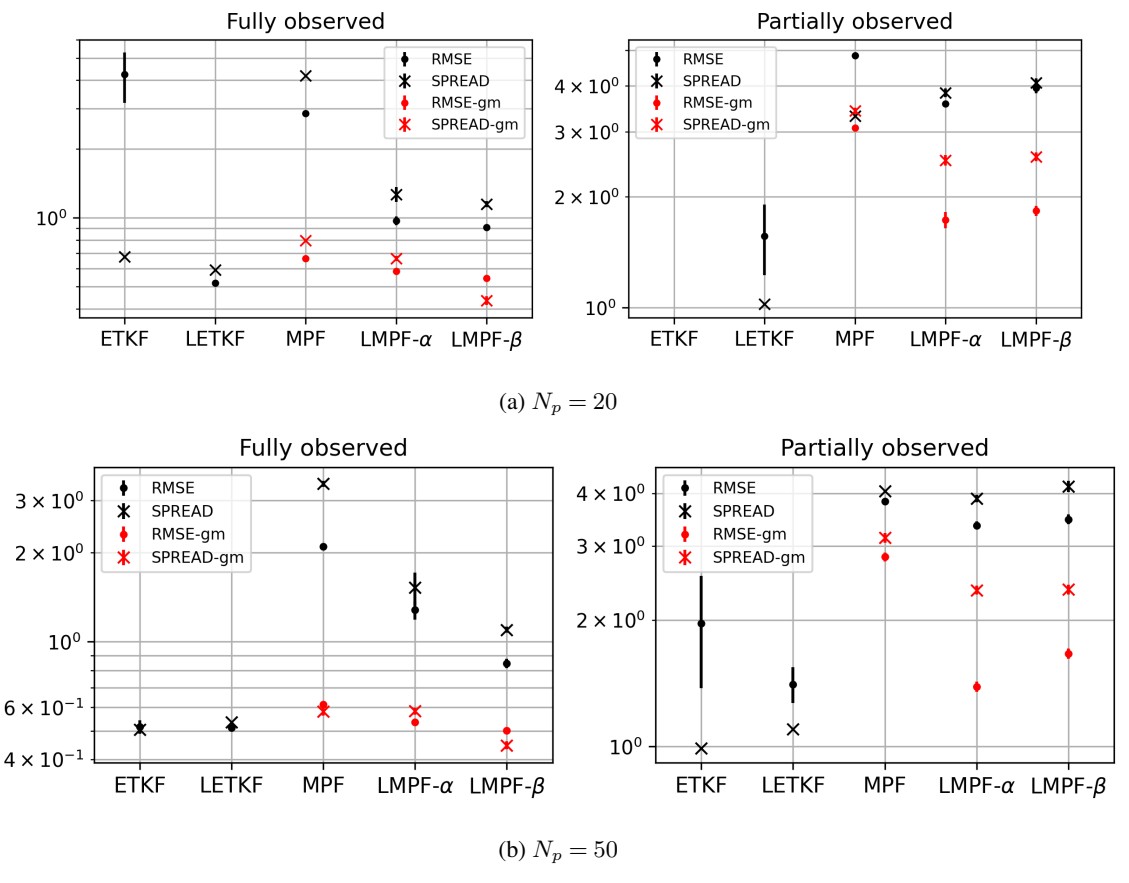

(a) $N_p = 20$

(b) $N_p = 50$

**Figure 4.** RMSE and spread in the logarithm operator for 20 (a) and 50 (b) particles, under both fully and partially observed scenarios.

For the 20-particle experiments in fig. 4a, both the ETKF and the Gaussian-prior global MPF achieved very high RMSE values in fully observed cases. In the partially observed case, the ETKF reached an RMSE>NoDA-RMSE. Meanwhile, the LETKF achieves excellent RMSE values with a closely matching spread in the fully observed case. In contrast, the Gaussian-priors filters exhibit the worst performance, while the Gaussian-mixture prior localized filters show good performance, comparable to the LETKF. While the LETKF shows the best mean RMSE, it exhibits large error bars due to its sensitivity to initial conditions. The Gaussian-mixture localized filters show slightly higher mean RMSE values, but these fall within the LETKF's error band and have much smaller error bars, resulting in comparable overall performance.

For 50 particles, fig. 4b, the ETKF successfully converges in the fully observed case, showing performance comparable to that of the LETKF. In this scenario, the Gaussian-mixture particle filters also demonstrate competitive results. In the partially observed scenario, ETKF and LETKF again show good results, but with large error bar in the case of ETKF. Meanwhile, the localized Gaussian-mixture filters have performance comparable to LETKF. In the case of LMPF-$\alpha$, the RMSE falls within the LETKF's error band. The MPF's in all its versions show smaller sensitivity to initial conditions, particularly in the partially observed case.

## 4.4 Sensitivity to the localization radius

The performance of localized particle filters is assessed by varying the radius of localization. This study is made on the linear fully observed case, and on the logarithm and partially observed case, the most non-Gaussian scenario. The number of particles used is 20 and only Gaussian-mixture prior densities are used in the MPFs.

Figure 5 shows the results of the linear experiment. The LETKF achieves a minimum RMSE at a localization radius of $\ell = 3$. This is the main reason why we selected this localization radius to conduct all localized experiments.

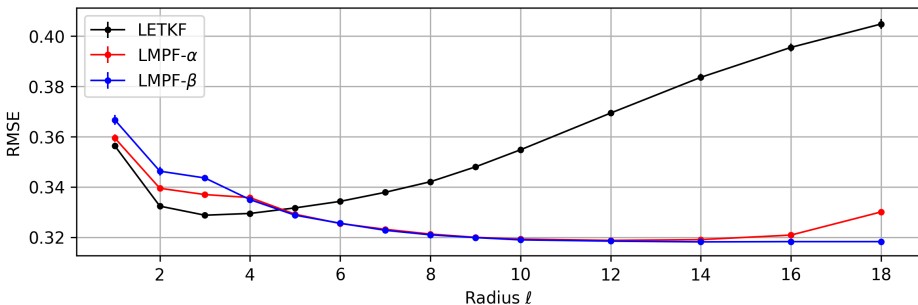

**Figure 5.** RMSE as a function of localization radius for the LMPFs and the LETKF for the linear and fully observed case with $N_p = 20$.

For radii greater than 4, the LETKF degrades more rapidly than LMPFs. The LMPFs tend to converge to the same RMSE performance as the global MPF when using a localization radius of 18. This suggests that for $N_p = 20$ the local algorithm benefits from incorporating distant covariances, even with reduced weights, to improve the estimation at each grid point.

LMPF-$\alpha$ exhibits a behavior similar to the $\beta$-case but results in slightly higher RMSE values and reaches a minimum around $\ell = 12$. However, the difference is small considering that the RMSE as a function of the localization radius is almost flat for that range of localization scales.

These results reflect the relationship between localization needs and the system's effective dimensionality relative to the ensemble size. For the 40-dimensional single Lorenz dynamics, the number of positive Lyapunov exponents is smaller than the ensemble size used in these experiments (20 particles). In the case of this surrogate model, using a forcing $f = 26$, the number of positive Lyapunov exponents calculated were around 16-17 with the parameterized forcing, $f^*$, and around 18-19 without it. With 20 particles exceeding the number of unstable directions, the ensemble in principle provides sufficient rank to capture the system's dynamics without requiring strong localization, explaining the optimal performance at larger radii.

To test this hypothesis, we conducted additional experiments with reduced ensemble size (10 particles), where the ensemble rank falls below the number of positive Lyapunov exponents.

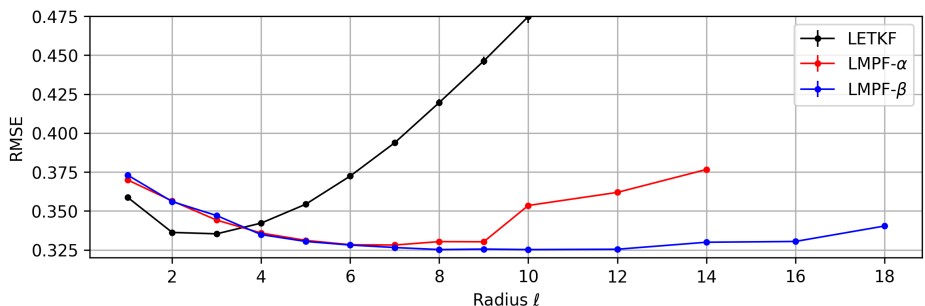

**Figure 6.** RMSE as a function of localization radius for the LMPFs and the LETKF for the linear and fully observed case with $N_p = 10$.

The results of this experiment are shown in fig. 6. The LETKF exhibits behavior similar to the 20-particle case. For the localized particle filters, a minimum value appears around $\ell = 9$. In the case of the LMPF-$\alpha$, the filter does not converge for localization radii greater than 14.

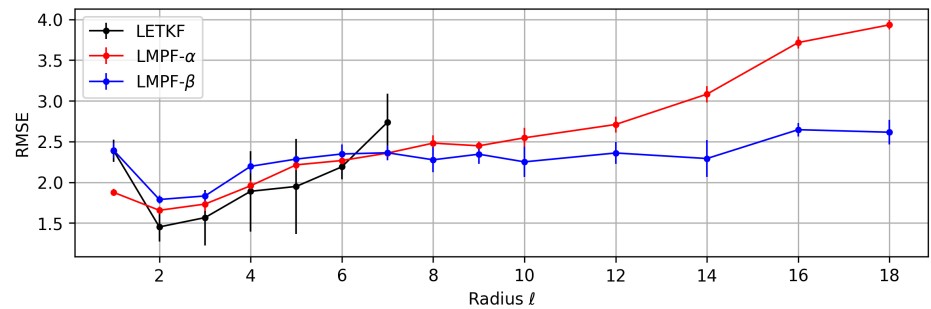

**Figure 7.** RMSE as a function of localization radius for the LMPFs and the LETKF for the logarithmic and partially observed case with $N_p = 20$.

We remind that the experiments are under the presence of model error. This affects the optimal localization radius; in particular, the LETKF has a longer optimal localization radius for twin perfect-model experiments. In realistic applications, the presence of model errors is also expected to affect long-range correlations. The MPF appears to behave more robustly to this effect.

Figure 7 shows the performance of the filters for the logarithmic and partially observed experiments for 20 particles. In this scenario, all the filters achieve a minimum RMSE around $\ell = 2, 3$. The LETKF shows more sensitivity to initial conditions than the localized filters; however, it achieves slightly lower RMSEs.

## 4.5 Sensitivity to the particle number

The two extreme experimental setups—fully observed with a linear observation operator, and partially observed with a logarithmic observation operator—are used to evaluate the sensitivity of performance to the number of particles. As in the previous subsection, only Gaussian-mixture prior densities are considered. The localization radius is fixed at $\ell = 3$, and the experiments are conducted with particle numbers of 5, 10, 20, 50, and 100.

Global MPF and LMPF-$\alpha$ demonstrate very good performance for small particle numbers in the linear experiment, fig. 8a. For larger particle numbers, both localized particle filters achieve excellent performance, comparable to that of the LETKF.

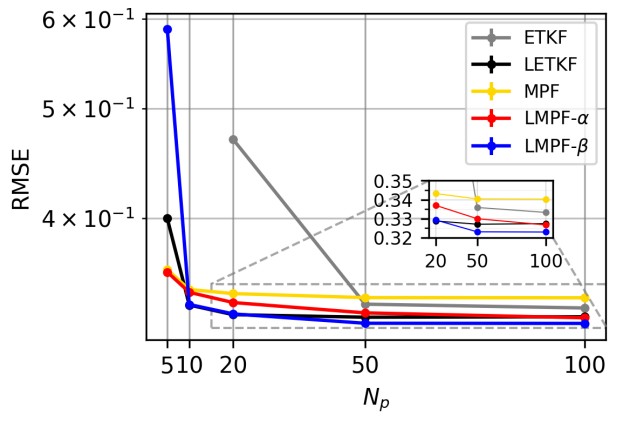

(a) Fully observed linear case.

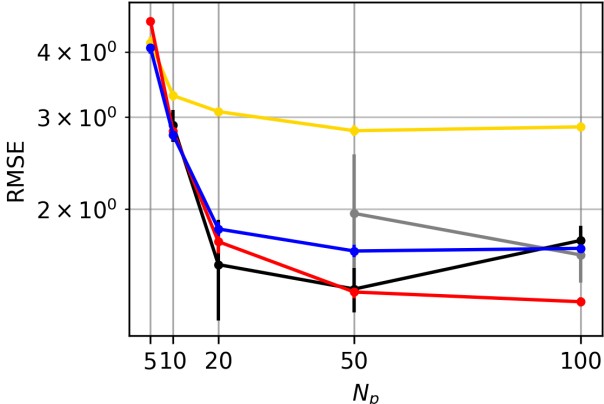

(b) Partially observed logartihmic case.

**Figure 8.** RMSE as a function of particle number for extreme cases.

In contrast, the results for the partially observed logarithmic case, fig. 8b, are unexpected. For a small number of particles, only Gaussian-mixture MPFs achieved RMSE less than NoDA-RMSE, although with a high RMSE. At larger particle numbers, LMPF-$\alpha$ achieves convergence with an accuracy greater than that of the Kalman filters. The performance of LMPF-$\beta$ is similar to Kalman filters.

The performance of the LETKF in this non-Gaussian experiment deteriorates for ensembles of 50 and 100 particles. A plausible explanation is that certain ensemble members diverge and fail to return to the Lorenz attractor, an effect that is found in

deterministic filters (Amezcua et al., 2012). The underlying reason for this behavior is that the data assimilation in deterministic Ensemble Kalman filters only scales the prior density without changing its shape. Consequently, if the prior ensemble contains outliers, they persist in the posterior density and can grow towards the next data assimilation cycle. LMPFs do not degrade in performance with increasing numbers of particles and appear to be unaffected by this issue, as the particle communication

during pseudo-time iterations modifies the prior's shape, effectively removing outliers. This issue in the LETKF could be mitigated through techniques such as applying random rotations to the analysis perturbations. However, for the purpose of this comparison, we implement a standard LETKF without additional enhancements to provide a consistent baseline against which to evaluate the proposed MPF and LMPFs methods. Interestingly, neither the MPF nor LMPFs exhibit this performance degradation with increasing ensemble size.

## 4.6 Sensitivity to large model error

In all previous experiments, a linear parameterization of small-scale effects was used. This results in a relatively small model error. To evaluate a large model error scenario, we neglect the linear term of the parameterization $f^*$ (table 2) and only use the external forcing $f(X_n) = 26$ in the surrogate model. As said before, NoDA-RMSE for the parametrized surrogate model is 6.78 and the spread is 6.55. For this large model error environment, the RMSE is 6.86 and the spread reaches 9.01.

Both Kalman and particle filters are tested for the linear and logarithmic observation operators, using $\ell = 3$ for the local filters and $N_p = 20$ particles. Again, we use the localization radius that is optimal for the LETKF, and the hyperparameters $\gamma$, $\xi$ and inflation are tuned for these cases. The results are displayed in fig. 9. In these large model error conditions, the ETKF achieved an RMSE>NoDA-RMSE.

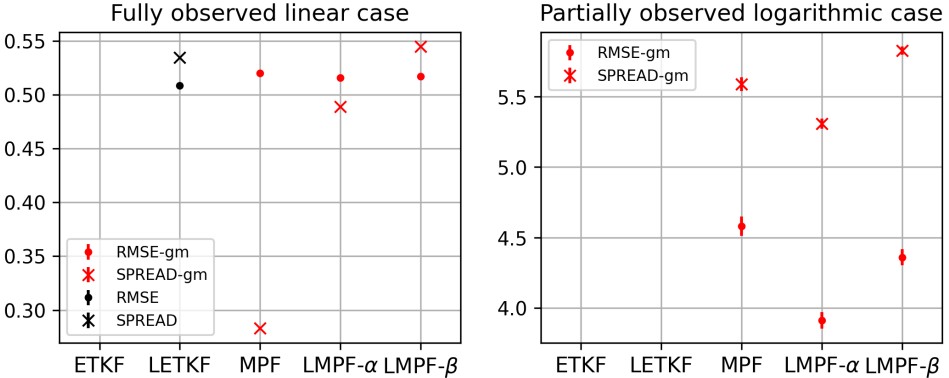

**Figure 9.** RMSE and spread for the large model error experiments using linear and fully observed cases and logarithm and partial observations with 20 particles and a localization radius of 3.

In the linear case, shown in fig. 9a, the Gaussian-mixture particle filters have a similar performance in terms of RMSE.
Nevertheless, the spread of the global MPF is strongly underestimated. Localized particle filters show higher spreads.

In the logarithmic scenario, the LETKF converged to an RMSE>NoDA-RMSE. In contrast, the MPF and its localized variants handle model error more effectively and perform better in this challenging case (fig. 9b). However, all three filters exhibit rather high RMSE values. Among the tested methods, LMPF-$\alpha$ achieves the best performance in this setup.

## 5  Conclusions

In this work, two localization schemes for the mapping particle filter were proposed. Both schemes are based on the hypothesis that distant observations do not impact the analysis, but their approaches differ. LMPF-$\alpha$ first calculates a global kernel covariance matrix and inverts it. Then, it performs local transformations at each pseudo-time step to obtain a global intermediate state vector in each step. Therefore, convergence is achieved globally. On the other hand, LMPF-$\beta$ applies the global algorithm in small regions, retaining the center value of each local analysis to obtain a smooth solution. Kernel covariance matrices are calculated in each small domain. Hence, each local analysis achieves convergence independently.

Both frameworks were tested in different setups and compared with the ETKF, LETKF, and the global MPF. In general, there is a clear positive impact when taking the prior probability density as a Gaussian mixture compared to a Gaussian prior density.

For both linear and non-linear operators, LMPF's improve estimation compared to their global version when a Gaussian prior is used and provide slightly better estimations when Gaussian mixtures are used. Furthermore, LMPF's provide better estimates compared to the ETKF and competitive performances against the LETKF.

In the linear case, LMPF's show very good estimations in terms of RMSE. In the squared case, Gaussian-mixture filters show very good estimations. Both Gaussian and non-Gaussian filters show poor spread representation, especially in partially observed scenarios. In the logarithmic case, Gaussian-mixture LMPF's provide competitive solutions against the LETKF. Again, the partially observed scenario degrades the performance of particle filters while Kalman filters are less affected. LMPFs present a very good performance in the logarithmic operator case under weak model error similar to LETKF.

When the number of particles varies, Gaussian-mixture MPF and LMPF-$\alpha$ show better estimates at low particle numbers. For the experiments with large model error the MPF and LMPF exhibit robust performances and successfully converge while ensemble based Kalman filters did not deal well with large model errors in the logarithmic experiment. However, it is important to highlight that all these experiments required brute-force optimization of two hyperparameters in Gaussian mixtures experiments which is computationally expensive.

The implementation of the particle filter for data assimilation in one-scale Lorenz model experiments represents an essential first step in validating our newly developed methodology. Working with simplified models provides a crucial foundation before advancing to more complex atmospheric forecast models, a direction which has already been explored by Hu et al (2024), suggesting that applying the proposed LMPF methodologies in large atmospheric models would also be feasible.

## Appendix A

The optimal hyperparameters for some of the the experiments are presented, where Kalman filters report the inflation factor, Gaussian-prior particle filters show the $\gamma$ parameter, and Gaussian-mixture filters display the $\gamma$ and $\xi$ parameters.

**Table A1.** Optimal parameters for the linear case.

|  | FO | | PO | |
|---|---|---|---|---|
|  | 20 | 50 | 20 | 50 |
| ETKF | 1.8 | 1.6 | 1.5 | 1.4 |
| LETKF | 1.49 | 1.46 | 1.32 | 1.3 |
| MPF-Gauss | 2000 | 250 | 750 | 150 |
| LMPF-$\alpha$-Gauss | 175 | 200 | 60 | 70 |
| LMPF-$\beta$-Gauss | 83 | 83 | 40 | 40 |
| MPF-GM | 34.0/1.0 | 24.0/0.6 | 19.0/0.75 | 10.0/0.75 |
| LMPF-$\alpha$-GM | 1.9/1.5 | 1.6/1.0 | 1.9/1.0 | 1.25/1.0 |
| LMPF-$\beta$-GM | 10.0/0.4 | 1.5/2.6 | 2.3/2.0 | 1.5/1.75 |

**Table A2.** Optimal parameters for the log case.

|  | FO | | PO | |
|---|---|---|---|---|
|  | 20 | 50 | 20 | 50 |
| ETKF | 2.8 | 1.4 | - | 1.3 |
| LETKF | 1.5 | 1.3 | 1.3 | 1.2 |
| MPF-Gauss | 250 | 115 | 500 | 100 |
| A-Gauss | 40 | 50 | 10 | 15 |
| B-Gauss | 20 | 25 | 10 | 12.5 |
| MPF-GM | 15.0/0.9 | 7.6/1.7 | 5.6/0.7 | 2.2/0.5 |
| A-GM | 2.0/0.7 | 1.3/0.5 | 0.4/0.2 | 0.25/0.2 |
| B-GM | 2.5/1.5 | 1.6/1.1 | 0.5/0.3 | 0.3/0.25 |

*Competing interests.* At least one of the (co-)authors is a member of the editorial board of Nonlinear Processes in Geophysics.

*Author contributions.* JG and MP participated in the conception of the ideas and in the design of the experiments. JG conducted the experiments. All the authors reviewed the results. JG and MP wrote the draft, all the authors made corrections and comments to the subsequent versions of the manuscript and approved the final version of the paper.

*Financial support.* National Agency for the Promotion of Science and Technology of Argentina (ANPCYT, Grant No. PICT 2019/3095) This research has been partially supported by the PREVENIR project implemented by the Japan International Cooperation Agency and the

Japan Science and Technology Agency under the Science and Technology Research Partnership for Sustainable Development Program.

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
