# Peer review of "Localization in the mapping particle filter"

_EGUsphere, 2025_

## Author Comment (AC1)

**Reviewer 1**

We are grateful to Professor Van Leeuwen for all the valuable comments and suggestions. We have revised the manuscript accordingly and provide our point-by-point responses below.

**Comment:**

*Abstract: It would be misleading to call the mapping particle filter a particle filter. This might seem a contradiction, but a particle filter is defined by likelihood weights, which are not present in a particle flow filter. A particle flow filter is an iterative ensemble method, perhaps better called an iterative MCMC method, and likelihood weights are not involved by construction. I suggest to remove the confusing statement referring to particle filters.*

**Response:**

In the Abstract, the statement regarding particle filters has been removed. We have changed to particle flow filter where it is pertinent. Additionally, some extra information about the MPF has been added in order to make it clearer.

**Comment:**

*32: A 4DVar is not a linear DA method, but an iterative nonlinear DA method. It is true that is can struggle when nonlinearity is strong, but that doesn't make it a linear DA method!*

**Response:** In line 32, we changed the term "linear" to "Gaussian" techniques.

**Comment:**

*42: Localization in particle filters was introduced in Bengtsson, T., Snyder, C. and Nychka, D. (2003) Toward a nonlinear ensemble filter for high-dimensional systems. Journal of Geophysical Research, 108, 8775–8785, and independently in van Leeuwen, P.J. (2003) Nonlinear ensemble data assimilation for the ocean. In seminar Recent Developments in Data Assimilation for Atmosphere and Ocean. 8–12 September 2003, ECMWF, Reading, UK.*

**Response:**

Suggested references were incorporated: line 42 includes a reference on localization in particle filters.

**Comment:**

*43: Jittering is used to rejuvenate particles after a tempering step. Please correct the wording.*

**Response:**

The correction regarding jittering was made in line 45, and the corresponding paragraph has been paraphrased following Dr. Farchi's recommendation.

**Comment:**

*76: A particle flow filter is not a special particle filter, because it does not use likelihood weighting, see point 1 above.*

**Response:**

The statement regarding particle filters has been removed in line 83.

**Comment:**

*61: Some recent literature is missing.*

*The stochastic version of the Particle Flow Filter is unbiased at any ensemble size, and solves many of the issues mentioned, see e.g. Gallego, V. and Insua, D. R. (2020)*

*Stochastic gradient mcmc with repulsive forces. arXiv preprint arXiv:1812.00071, https://arxiv.org/abs/1812.00071; Leviyev, A., Chen, J., Wang, Y., Ghattas, O. and Zimmerman, A. (2022)*

*A stochastic stein variational newton method. arXiv:2204.09039.*
*https://doi.org/10.48550/arXiv.2204.09039; Ma, Y.-A., Chen, T. and Fox, E. (2015)*

*A complete recipe for stochastic gradient mcmc. In Advances in Neural Information Processing Systems (eds. C. Cortes, N. Lawrence, D. Lee, M. Sugiyama and R. Garnett), vol. 28. Curran Associates, Inc. https://doi.org/10.48550/arXiv.1506.04696.*

**Response:**

Line 63 includes the suggested studies addressing SVGD-related problems.

**Comment:** *79: The description of Hu and Van Leeuwen is incorrect. They use a preconditioning matrix in the flow to speed up convergence, and they choose a localizated prior covariance for this matrix. The result is that the prior covariance matrix is cancelled by this precondition matrix. Nothing special is done to the likelihood. Note also that the preconditioning matrix does not have to be very accurate, a rough localization will do. This is a distinct difference with LEnKFs, in which accurate localization is crucial. (PS the method is also applied to a full atmospheric model in Hu, C-C , P.J. van Leeuwen , J. L. Anderson (2024) An implementation of the particle flow filter in an atmospheric model, Monthly Weather Rev., doi: 10.1175/MWR-D-24-0006.1.)*

**Response:**

In line 87, we corrected the description of the method by Hu and Van Leeuwen and added the suggested reference regarding the implementation of the methodology in an atmospheric model.

**Comment:**

*93: Subrahmanya et al also minimize the KL divergence, but formulate the flow field using the FP equation, and then propose approximations to come up with solutions for high-dimensional systems. The main difference with Pulido and Van Leeuwen is that they do not use a RKHS.*

**Response:**

Line 107 incorporates the comments regarding Subrahmanya's work.

**Comment:**

*Eq. (5): This flow, together with Eq. (4) does not conserve the physical dimension of the state and hence is inconsistent. We did this wrong in 2019, apologies. Assume the state contains temperature measured in K. Then the flow has physical dimension $K^{-1}$ (assuming the kernel has no physical dimension, such as the one used in Eq. (6)), which is inconsistent with Eq. (4). The preconditioning with a covariance matrix of the state as explored by Hu and Van Leeuwen is one way to solve this issue.*

**Response:**

Equation (5) was modified in order to include the corresponding dimensions of the velocity. Accordingly, eq. (14) and eq. (17) were also modified. Line 145 explains this factor is incorporated in the learning parameter.

**Comment:**

*147: Note that if the model is stochastic with additive Gaussian errors, Eq. (9) is not an approximation, see Pulido and Van Leeuwen. It might be good to point that out.*

**Response:**

Following the reviewer's suggestion, we added a note in line 171 to indicate that if the model is stochastic with additive Gaussian errors, eq. (11) is not an approximation.

**Comment:**

*160: The beta localization comes close to the methodology implemented in Hu et al. Monthly Weather Rev., doi: 10.1175/MWR-D-24-0006.1.*

**Response:**

The resemblance between the $\beta$-algorithm and the methodology of Hu et al. (2024) is mentioned in line 200.

**Comment:**

*Eq 10: note difference between independence and uncorrelated. Perhaps rephrase sentence above this.*

**Response:**

We have clarified the statement above eq. (13) in line 205. In the revised version, we removed the reference to "independence" .

**Comment:**

*Eq. 11: Is there a reason to change the nabla notation?*

**Response:**

In eq. (14) we kept the l-component of the eq. (5). Nevertheless, there was a typo. The l-component of $\nabla_{\mathbf{x}}$ is the partial derivative $\partial_{x_l}$ and we wrote $\partial_{\tilde{x}_l}$. This typo has been corrected. We have also defined the $\partial_{x_l}$ operator.

**Comment:**

*205: Please finish the sentence.*

**Response:**

The incomplete sentence at line 249 was removed.

**Comment:**

*233: In the beta algorithm, assume one grid point is completely updated, and we move to the next point. Will that point use the updated first grid point value, or the original first grid point value? The former will lead to smoother global fields, but makes the result depend on the order in which the grid points are updated. One could imagine a mixture between alpha and beta, where beta is used over the whole filed at each iteration. Hu et al. Monthly Weather Rev., doi: 10.1175/MWR-D-24-0006.1 use another local updating scheme. It might be something to discuss; it would help future users of these methods.*

**Response:**

In the $\beta$ approach, the update of each grid point is performed with respect to the background state, and not with the sequentially updated values of previously visited points. Therefore, the results do not depend on the order in which the grid points are visited, this is clarified in line 276.

**Comment:**

*Eq. (23): what does the index LS mean? I assume Large Scale?*

**Response:**

In line 319 we added that the index LS stands for large-scale.

**Comment:**

*263: Notation: M means something different in section 3.0 compared to here.*

**Response:**

The notation for the true and surrogate models has been clarified: $\mathcal{M}^t$ denotes the true model and $\mathcal{M}^{su}$ the surrogate model (line 292).

**Comment:**

*324: Particle filters $\rightarrow$ particle flow filters*

**Response:** In line 390, "particle filters" has been replaced by "particle flow filter".

**Comment:**

*Experiments: It doesn't make sense to tune methods on lowest RMSE because ensemble spread is as important. For instance, in operational weather prediction centers both are optimized. If not corrected in the experiments, please provide some discussion.*

**Response:**

We completely agree with the reviewer. In line 375, we added a discussion on the relationship between RMSE and ensemble spread. While our hyperparameter optimization focuses on minimizing RMSE, we also examine how the ensemble spread behaves under these conditions. Specifically, we investigate whether the filter maintains a realistic representation of the spread when RMSE is minimized. Our motivation is to examine whether the methodologies produce a reasonable representation of the spread at their lowest RMSE without explicitly tuning for it. This was indeed the case for some of the experiments and methodologies, demonstrating that reasonable spread can emerge naturally without direct optimization.

**Comment:**

*Experiments: Can the authors provide a rule of thumb for choosing the hyperparameters?*

**Response:**

Line 379 clarifies that hyperparameter optimization is performed by brute force. The RMSE dependence on the hyperparameters is nontrivial and non-intuitive, which prevents the definition of a simple rule of thumb. There is an Appendix with the optimal hyperparameters for some of the experiments showing this behaviour. This discussion has been added to the revised manuscript.

**Comment:**

*Experiments: Can the authors provide a rough comparison of computational costs compared to the LETKF?*

**Response:** The computational complexity of the three algorithms is given in lines 176, 241, and 284.

**Comment:**

*456: I would not say that the LMPF are "highly competitive" compared to an LETKF. They tend to be worse, and with similar performance at best. The spread is typically underestimated, which is not a good sign.*

**Response:**

The statement regarding comparison with the LETKF in line 537 has been softened from "highly competitive" to "competitive". We note that the performance of LMPF in the extended experiments with different seeds shows some improvements in terms of RMSE and spread, and both LMPFs have a quite robust performance across experiments.

**Comment:**

*469: Convergence results of the MPF rely on keeping the kernel a fixed function of its two arguments, and changing the covariance matrix does not fulfill this criterion. "this matrix must evolve with pseudo time" is incorrect. Please change the text and mention this issue.*

**Response:** The paragraph on pseudo-time evolution of the kernel covariance matrix has been removed.

**Comment:**

*474: A step in this direction is Hu et al Monthly Weather Rev., doi: 10.1175/MWR-D-24-0006.1.*

**Response:**

Finally the reference to Hu et al. (2024) has been added in line 550.

**Reviewer 2**

We thank the reviewer, Dr. Farchi, very much for all your comments. We address each point below.

**Comment:**

*From what I understood, the introduction of $\tilde{x}$ instead of $x$ is not fundamental (meaning that the expressions would still be valid by replacing $\tilde{x}$ by $x$) but practical (because of the localisation, many entries of the covariance matrices are zero and hence the matrix-vector products can be computed using only a subset of the state). If this is indeed the case, I think that this should be explained at the end of the section. The authors could perhaps even consider presenting a version without $\tilde{x}$ and only mention that the algebra can be easily optimised by selecting the non-zero entries in the matrices and vectors at the end of the section. I have the feeling that it would make the section easier to understand.*

**Response:**

Good point from the reviewer. Indeed, the assumption of the $\tilde{\mathbf{x}}$ local variable is not essential; we could also start by assuming that each of the involved (state-space) covariance matrices is l-block diagonal. However, we think that local variables are certainly useful for interpretation and for avoiding some notation ambiguities, particularly when calculating the terms that compose eq. (17). As the reviewer noted, if we assume l-block diagonal matrices $\Lambda_l$ in the kernel, the local kernel is the same whether we use the local state vector or the global one, because of the zeros in the off-diagonal blocs of $\Lambda_l$ corresponding to variables distant from $x_l$. We think the $\tilde{\mathbf{x}}$ local variable assumption is more straightforward: once it is adopted, the local versions of the kernel, prior density, and likelihood function are naturally established. On the other hand, the assumption of l-block diagonal matrices would need to be applied to each matrix individually and particular notation should be introduced to avoid ambiguities.

We have added a discussion of this point in the manuscript in line 237. We thank the reviewer for noting this subtle point.

We have also noted, for clarification of the point, that we can calculate a local version of the prior density in eq. (11) using a "localized global approach" where $\mathbf{x}_k$ is a global state vector and $\mathbf{Q}_k$ consists of banded/Gaspari-Cohn/ localized matrices, and then use the local prior gradient. This global-to-local prior density approach in preliminary experiments led to suboptimal results, with the explicitly local calculation of the prior density that we showed having better performance.

**Comment:** *In the section, several modifications to the original equations are presented, but without writing the modified equations. While this helps keeping the manuscript short and focused, it makes it harder to follow. Therefore, I think that it would make sense to present the finalised expressions, perhaps in an appendix, in one of the two cases for the prior density (Gaussian or Gaussian mixture).*

**Response:**

To address this concern and improve clarity, we have incorporated the explicit velocity term derivation from eq. (5) in eq. (12). This addition presents the final expression for the specific case of a radial basis function kernel combined with a Gaussian prior, making the mathematical development more transparent and easier to follow.

**Comment:**

*I think that a discussion about the algorithmic complexity of the algorithm is missing.*

**Response:**

The computational complexity of the three algorithms is given in lines 176, 241, and 284.

**Comment:**

*Somewhere in the text, it should be mentioned that beta-localisation can only be applied in the case of local observations (i.e. observations which have a well-defined location in physical space).*

**Response:**

This point has been added in line 259.

**Comment:**

*It would be interesting before section 4 to make a summary, for each filter of the tuning parameters. Unless I missed it, it is not specified whether the parameters are tuned separately for each experiment or once and for all experiments. Furthermore, I don't understand why the localisation radius is not included in the set of tuning parameters. In my opinion, using a fixed value of localisation radius favours one algorithm over the others in a quite arbitrary way.*

**Response:**

A table summarizing the optimal hyperparameters for each experiment has been added in an Appendix.

We acknowledge that the radius of 3 is optimal specifically for the LETKF with 20 ensemble members in the linear case. The choice of a fixed localization radius for all methods was deliberate to ensure a fair comparison with LETKF. In preliminary experiments where we changed the localization radius, we detected that the MPF has systematically a larger localization radius than LETKF. Because the competitive results obtained with the LMPF compared to the LETKF could be (wrongly) attributed to the differences in the localization radius, we decided to keep the same localization radius across all the experiments and to show explicitly in an experiment the performance dependence of the methods on the localization radius.

We have clarified this methodological choice in the revised manuscript in lines 416-419 and acknowledged that individual optimization for each ensemble size and method would likely yield different optimal radii. Importantly, the LMPF's show better performance at longer radii as demonstrated in our sensitivity analysis, indicating that our fixed radius choice does not bias the comparison in favor of the methodology we are introducing in the work.

**Comment:**

*2.2k cycles for the Lorenz system is really small and makes the results really sensitive to the specific choice of random seed. I would highly recommend to use at least 10k cycles (100k would be better) and to repeat each experiments N times with N different random seeds (from which you could get error bars for the results). This sensibility is really visible in some figures which look "noisy" (eg fig 7), and error bars could help distinguish what differences are significant or not.*

**Response:**

Thanks for this observation. We have conducted all the experiments again for 10k cycles, using 1k as spin-up, and we have repeated each experiment 10 times with 10 different seeds. We thank Dr. Farchi; these modifications have enhanced the results.

**Comment:**

*Sometimes, you mention that one or the other of the filters did not converge. Could you be more explicit by what you mean here, for both the (L)MPF and for the (L)ETKF?*

**Response:**

Regarding this comment, we have corrected all the unspecific results, highlighting the difference between the case when the final RMSE was greater than the climatologic error, and the case when the experiment was cut due to numeric divergence.

**Comment:**

*Finally, even if it is already a good step to make a comparison with the optimally tuned (L)ETKF, it should be noted that the (L)ETKF is not designed to handle non-linearity and model error. Therefore, in the non-linear cases, the EnKF cannot really be considered state-of-the-art and should in principle be replaced by iterative EnKFs, and in all cases, there should be a treatment for model error (even a basic one) in the EnKFs. I can understand that making a full comparison could be out of scope of the present work, but I think that these points should be at*

*least mentioned*

**Response:**

We appreciate this important observation. While we acknowledge that iterative EnKFs and methods with explicit model error treatment would provide more sophisticated benchmarks, our objective is to evaluate the MPF against widely-used classical filters rather than state-of-the-art methods. The LETKF and ETKF serve as representative baseline algorithms that are computationally efficient and commonly employed in operational settings. We have added this clarification to the results section in line 362.

**Comment:**

*L4: "the Stein variational gradient descents" → "the Stein variational gradient descent"?*

**Response:**

This typo has been corrected in line 4.

**Comment:**

*L 5: "aiming to minimize the Kullback-Leibler divergence" I find this sentence not entirely clear: who is "aiming to" and between what and what is the KL divergence computed?*

**Response:**

The divergence minimization is between the prior density and the posterior. This is clarified in line 6.

**Comment:**

*L 9: "Gaussian and Gaussian mixtures are evaluated as a prior density." This formulation seems a bit weird. In addition, I am not entirely sure that this information is essential in the abstract.*

**Response:**

Regarding the suggested comment, we decided to omit the prior density information in this paragraph.

**Comment:**

*L 32: "such as four-dimensional variational" 4D-Var is not a linear data assimilation method! Here (and in the following sentences) you are mixing non-Gaussianity and non-linearity. It is true that non-linearity does induce non-Gaussianity, but the two effects should remain distinguished in my opinion.*

**Response:**

In line 32, we changed "linear" to "Gaussian".

**Comment:**

*L 42: I think that there is a typo in the name of Poterjoy.*

**Response:**

Yes, thanks.

**Comment:**

*L 42-44: Localisation, tempering, and jittering techniques have different objectives. Therefore, I find it a bit weird to see them in the same sentence (even though it is true that they all ultimately aim at making the particle filter work).*

**Response:**

The correction regarding jittering was made in line 45, and the corresponding paragraph has been paraphrased.

**Comment:**

*L 46: "The MPF is a novel data assimilation approach" While I agree that the MPF is more recent than other DA algorithms, it has already been introduced 6 years ago, based on the SVGD proposed 9 years ago. Can we still*

*say it is "novel"?*

**Response:**

The description of the MPF as "novel" has been softened in line 48.

**Comment:**

*L 79-83: Can you briefly describe here how your method differs from that of Hu and van Leeuwen, 2021?*

**Response:**

This comment was addressed in lines 188-195 where the main differences between our approach and that of Hu et al. (2021) were added before explaining the localization schemes in detail. In this work, we start from the localization assumption and apply it consistently, in terms of the localization radius, to the prior density, likelihood and kernel function.

**Comment:**

*L 84-86: The outline is missing at the end of the introduction.*

**Response:**

We have added an outline at the end of the introduction in line 99.

**Comment:**

*L 108: "that sample" → "that samples"*

**Response:**

The typo has been corrected in line 122.

**Comment:**

*L 121: please avoid nested parentheses in citations.*

*Response:*

This suggestion was addressed and the nested parentheses were removed in line 135.

**Comment:**

*L 124: "Kullback-Leibler Divergence DKL" between what and what?*

**Response:** The KL divergence is between the target posterior density and the intermediate density. This clarification has been added in line 139.

**Comment:**

*Before Eqs. (8) and (9) It would be nice to give the expression of the associated priors, no*

**Response:**

The expressions associated were added in eq. (8) and eq. (10).

**Comment:** *L 158: "global banded prior covariance matrix" wouldn't it be more appropriate to say "localised prior covariance matrix"?*

**Response:**

Indeed. This has been corrected in line 197.

**Comment:**

*L 194: A punctuation sign is missing at the end of the sentence.*

**Result:**

It has been added in line 233.

**Comment:**

*L 205: the sentence is not finished.*

**Response:**

The incomplete sentence in line 249 was removed.

**Comment:**

*Table 1: "Long-scale" → "Large-scale"?*

**Response:**

The error has been modified in Table 1.

**Comment:**

*Eq. 22: $N_{SU}$ should be $N_{SS}$, right? Also it would be good to define the acronyms LS and SS.*

**Response:**

$N_{SU}$ is the dimension of the surrogate model that can only represent the large-scale variables of the nature model. So in order to be consistent, $N_{SU}$ must be equal to $N_{LS}$. This is clarified in line 311. Also in lines 302 acronyms LS and SS are defined.

**Comment:**

*L 270: As it is, R corresponds to a variance (and not a standard deviation). I would therefore advise to use a symbol "squared" (eg $\sigma^2$...)*

**Response:**

This notation has been corrected in Eq. 24 and consequently in line 325.

**Comment:**

*L 279-281: Why not using the surrogate model to initialise the particles? This would make the algorithm completely independent of the true model.*

**Response:**

We repeated the experiments using an initial ensemble from the surrogate model in order to be independent of the true model. This change is reflected in the results and in line 333.

**Comment:**

*L 289-291: It is in my opinion not clear here that you are referring to the experiments in section 4.4.*

**Response:**

We have clarified this by specifying which experimental setups are being referenced. This is corrected in lines 342-348.

**Comment:**

*L 307: please specify between what and what you compute the RMSE (analysis ensemble mean and truth I imagine).*

**Response:**

Indeed, it is between analysis ensemble mean and truth. This is clarified in line 365.

**Comment:**

*Figure 1 is nice and beautiful, but I am not sure that it is absolutely necessary. If you chose to keep it, I would strongly recommend to replace the colorbar "jet" by a perceptually uniform colormap*

**Response:**

We decided to keep the figure as it is useful to show the reader the non-trivial behaviour between the hyperparameters and the minimum RMSE. The coupling between the two hyperparameters shown in the figure is the reason why we decided a 2D optimization. We changed the figure to a perceptually uniform colormap as suggested (magma_r).

**Comment:**

*L 349: "which is optimum for the ensemble Kalman filter." This is true only for 20 ensemble members. For 50 ensemble members, given the fact that the ETKF performs approximately on par with the LETKF, I suspect that the optimal radius would be the largest one.*

**Response:**

This was clarified and addressed in lines 416-419.

**Comment:**

*L 394: I find it a bit deceiving to speak of "global case" when l=18. Indeed even if all the variables are included in the local domains, the Gaspari-Cohn tapering does make localisation different from the global case.*

**Response:**

The reviewer is correct that l=18 with Gaspari-Cohn tapering is algorithmically different from the truly global case. What we intended to explain is that the RMSE performance of the LMPF with l=18 converges to similar values as the global MPF, not that they are algorithmically equivalent.

We have revised the text to clarify in lines 469-471 that we are referring to performance convergence rather than algorithmic equivalence.

**Comment:**

*L 401: Can you give the number of positive (and neutral) Lyapunov exponents.*

**Response:** We calculated approximately 16-17 positive Lyapunov exponents (with parameterized forcing) and 18-19 positive Lyapunov exponents (without parameterization) for the 40-dimensional Lorenz system with $f = 26$. This information has been added in line 478.

**Comment:**

*Figure 8 could be shown in log-log scale.*

**Response:**

We have updated fig. 8 to use a log-log scale as suggested. Additionally, we have added a zoomed inset to better highlight the subtle differences between the methods in the low RMSE region.

**Comment:**

*L 425-428: Usually, this issue can be solve by applying random rotations to the analysis perturbations.*

**Response:** We have revised the text to acknowledge that the LETKF performance degradation could indeed be addressed through techniques such as random rotations to the analysis perturbations. However, for the purpose of this study, we deliberately use a standard LETKF implementation to provide a consistent and widely-used baseline for comparison with our proposed methods. This has been explained in lines 507-510.

**Comment:**

*L 452: "requires the prior density function to be declared beforehand" Do you mean that the parametrisation of the prior density should be chosen beforehand? But isn't it the case for most DA methods?*

**Response:**

Indeed, the reviewer is correct. Most DA methods require some form of prior specification. This phrasing was misleading and has been removed in line 532.

**Comment:**

*L 469: "this matrix must evolve with pseudo time" I am not sure to understand what is meant here.*

**Response:**

The confusing paragraph on pseudo-time evolution of the kernel covariance matrix has been removed.

**Comment:**

*About the name of the low-order model: the model known as "Lorenz 1996" (Lorenz and Emanuel, 1998, https://doi.org/10.1175/1520-0469(1998)055%3C0399:OSFSWO%3E2.0.CO;2) does not include a two-scale version. The two scale version was only proposed later on (Lorenz 2005, http://dx.doi.org/10.1175/JAS3430.1) and should be rigorously named "Lorenz 2005-III" (as it is the third model presented in this article). A good compromise could be to use the following names: "one-scale Lorenz model" (referring to the Lorenz 1996 model) and "two-scale Lorenz model" (referring to the Lorenz 2005-III model). Also, please do not forget the citations.*

**Response:**

The two-scale Lorenz model citation has been introduced in line 299 and the one-scale model citation in line 306. Furthermore, all references to these models throughout the manuscript have been appropriately modified following the reviewer's suggestion to use "one-scale Lorenz model" and "two-scale Lorenz model" respectively.

**Comment:**

*The format of cross-references to equations, algorithms, etc. are inconsistent throughout the manuscript, which makes it harder to follow. I strongly recommend the use of the latex package "cleveref" which automatically handles them.*

**Response:**

We have now implemented the cleveref package as suggested and have gone through the entire manuscript to ensure all cross-references are consistently formatted.

---

## Editor Decision (ED1)

Both referees, who are the same ones as for the first version of the paper, recommend acceptance of the paper, subject to minor revisions and technical corrections. I follow their advice, adding that I have myself below a number of editing suggestions.

I ask the authors to revise their paper along the referees' comments and suggestions, as well as along mine. I ask them to give a point-to-point response to all of these comments and suggestions. In case they do not agree with a particular comment or decide not to follow a particular suggestion, they must state precisely their reasons for that.

My editor's suggestions

1. Ll. 42-43, … *such as localization, first introduced in (Bengtsson et al., 2003) and independently in van Leeuwen (2003)*. You then mention (l. 75) earlier references to localization (in the context of EnKF, but that may be confusing).

I suggest you change the text above to … *such as localization, first introduced for particle filters, and independently, in (Bengtsson et al., 2003) and van Leeuwen (2003)*.

2. Ll. 62-63, sentence starting *To address this lack* … I suggest *To improve its convergence properties, Ba et al. (2022) proposed alternative formulations of SVGD*

3. L. 82, *While **that** work* …  is more appropriate here (compare with *This work* … in l. 47)

4. L. 125, define superscripts *f(j)* and *a(j)*

5. Ll. 200-201, word missing ? …. *in Hunt et al. (2007) **and** has some resemblance ...*

6. Ll. 205 and 214 (and maybe elsewhere). From what I understand, it is not a question of correlation, but of statistical dependence. I suggest to write on l. 205 *we assume that the variables located outside of a neighborhood $C_l$ of $x_l$ are statistically independent of $x_l$.*

7. L. 219, $\Gamma_l$ o $\Sigma$, I understand o denotes the Schur product ?

8. 219-220,  … *around l with only one's and zeros, as in eq. (15) below*

9. L. 478, the number given there (16-17, 18-19) must be the number of positive Lyapunov exponents, not their value. Say it more precisely.

---

## Author Response (AR2)

**Reviewer 1**

We are grateful to Dr. Van Leeuwen for all the valuable comments and suggestions. We have revised the manuscript accordingly and provide our point-by-point responses below.

**Comment:**

*Line 6: I would suggest a small change to emphasize that the KL divergence is minimized at each iteration between the pdf at that iteration and the posterior pdf, starting at the first iteration with the prior pdf.*

**Response:**

In line 5, we have incorporated the reviewer's suggestion to clarify the relevant details.

**Comment:**

*Line 32: This is still not correct as the likelihood can be non-Gaussian. Please be accurate. Note also the word 'also' in line 35. This is simply not correct.*

**Response:** The statements have been removed from lines 33 and 36.

**Comment:**

*Introduction: I would strongly suggest to split the discussion of the particle filter and the particle flow filter. They are completely different methods, both in philosophy and in implementation. The authors keep mixing them in their discussion, which is not helpful to the community. My suggestion is to first discuss the need for nonlinearity, then discuss the particle filters, then their issues and potential solutions (and I notice proposal densities are not discussed at all, see e.g. the Particle filter review in Van Leeuwen et al. 2019 for a more complete discussion, but there are many other sources). Then discuss another solution, the particle flow filters, and the connection with MCMC methods for the stochastic variants.*

**Response:**

We have restructured the introduction to clearly separate the discussion of particle filters and particle flow filters. The revised introduction now discusses proposal densities (lines 43-46), introduces particle flow filters as an alternative approach with their connection to MCMC methods (lines 52-56), and reorders the presentation to better distinguish these methodologies.

**Comment:**

*Line 213: "with respect to the l variable."*

**Response:**

We have made this grammatical correction in line 227.

**Comment:**

*Line 510: The underlying reason that the Kalman filters show this behavior is that the data assimilation only scales the prior, it does not change its shape. Hence, if the prior has one or two outliers, these outliers will remain in the posterior, and can grow towards the next data assimilation time, etc. The PFF do change the shape of the prior due to the communication among the particles during the iterations. The authors might want to add something like this.*

**Response:**

This suggestion has been addressed in line 521.

**Reviewer 2**

We greatly appreciate Dr. Farchi's thoughtful follow-up comments and careful review of our revised manuscript. Our detailed responses are provided below.

**Comment:** *L 27-28: repetition "particularly" / "particular" in the first sentence.*

**Response:**

The repeated use of "particularly" and "particular" has been corrected by replacing "particularly" with "especially" in line 27.

**Comment:** *L31: "the challenge" → be more precise: which challenge?*

**Response:**

The reference to "the challenge", in line 31, has been clarified as relating to the representation of enhanced non-linear and non-Gaussian behaviour.

**Comment:**

*L32: I think that this statement is still deceiving. The assumption of Gaussian errors in 4D-Var is not fundamental, and made for practical reasons, contrary to the Kalman filter which is by construction a Gaussian method. Perhaps you can simply remove 4D-Var from this sentence, without changing your message?*

**Response:**

The statement has been removed from line 33.

**Comment:**

*L 43-45: You could perhaps slightly reformulate the paragraph. In particular, avoid starting two consecutive sentences with "Other".*

**Response:**

The paragraph has been slightly reformulated by removing the first "Other" and replacing the second one with "Further" to improve readability (line 48).

**Comment:**

*L 45: This is a detail, but "jittering" is actually the same technique as what other authors (including myself) call "regularisation" and is not only used after tempering steps, but also in more "standard" implementations of the PF after (or before) resampling.*

**Response:**

We modified the sentence to indicate that jittering (or regularisation) is not restricted to tempering steps, but can also be applied after resampling (line 50).

**Comment:**

*Eq. 5: Introducing a constant "C" to have consistent physical dimensions looks like a quick patch that could potentially hide a more serious issue. That being, reading the other reviewer comment, I have the feeling that fixing this issue seems more complex than one can think of. For the present work, I think that it would be perhaps more appropriate to recognise that there is some inconsistency, and that more research is needed to solve it. (But this is only my opinion)*

**Response:** We have now clarified that this is not an inconsistency but that this work uses nondimensionalized variables as in previous references, (line 153). If one wants to work with dimensionalized variables a diffusion coefficient must be included in the formulation. The diffusion coefficient value in that case is related to the convergence rate but it has no physical meaning because the convergence is in pseudo-time (at a fixed physical time).

**Comment:**

*L 175-180: perhaps you could give a little bit more detail and briefly explain to what each term in L. 177 and 179 correspond? Also, I strongly advise to number all equations.*

**Response:**

We have added clarifications explaining the computational complexity terms in line 190. Following the reviewer's advice, we have also numbered all equations throughout the manuscript.

**Comment:**

*L 417: "the LMPF's" → "the LMPFs"?*

**Response:**

We corrected the plural of LMPF and applied the same change consistently throughout the manuscript.

**Editor**

We thank the Editor, Dr. Talagrand, for the suggestions and careful evaluation of our manuscript. We have revised the paper in accordance with these remarks, and our responses are detailed below.

**Comment:** *Ll. 42-43, "... such as localization, first introduced in (Bengtsson et al., 2003) and independently in van Leeuwen (2003)". You then mention (l. 75) earlier references to localization (in the context of EnKF, but that may be confusing). I suggest you change the text above to "... such as localization, first introduced for particle filters, and independently, in (Bengtsson et al., 2003) and van Leeuwen (2003)".*

**Response:**

The sentence in lines 48–48 has been modified according to the editor's suggestion.

**Comment:** *Ll. 62-63, sentence starting "To address this lack ..." I suggest "To improve its convergence properties, Ba et al. (2022) proposed alternative formulations of SVGD"*

**Response:**

The text in line 71 has been revised as suggested by the editor.

**Comment:**

*L. 82, "While **that** work ..." is more appropriate here (compare with "This work ..." in l.47)*

**Response:**

The sentence in line 91 has been changed according to the editor's instructions.

**Comment:**

*L. 125, define superscripts $f(j)$ and $a(j)$*

**Response:**

This point is clarified in line 133.

**Comment:**

*Ll. 200-201, word missing ? "... in Hunt et al. (2007) **and** has some resemblance ..."*

**Response:**

The sentence in line 214 has been corrected according to the editor's instructions.

**Comment:**

*Ll. 205 and 214 (and maybe elsewhere). From what I understand, it is not a question of correlation, but of statistical dependence. I suggest to write on l. 205 "we assume that the variables located outside of a neighborhood $C_l$ of $x_l$ are statistically independent of $x_l$".*

**Response:**

We have addressed this suggestion, and the text has been modified in line 218.

**Comment:**

*L. 219, $\Gamma_l \circ \Sigma$ , I understand $\circ$ denotes the Schur product ?*

**Response:**

This point has been clarified in line 232.

**Comment:**

*L.219-220, "... around l with only one's and zeros, as in eq. (15) below"*

**Response:**

The text in line 234 has been revised as suggested by the editor.

**Comment:**

*L. 478, the number given there (16-17, 18-19) must be the number of positive Lyapunov exponents, not their value. Say it more precisely.*

**Response:**

This point has been clarified in line 492.

---

## Editor Decision (ED2)

1. There remain acronyms that have not been explicitly expanded. For instance

- ETKF (first introduced in abstract l. 14 and in text l. 378)
- LETKF (ll. 14 and 37)

Concerning the latter, the authors mention l. 214 the paper by Hunt et al. (2007) without saying that it is precisely in that paper that the LETKF was first introduced (the connection is mentioned l. 377, but it must be mentioned earlier).

Please check that all acronyms are fully developed the first time they are used (both in the abstract and in the main text).

2. L. 333, … *10,000 cycle times*. How long is a cycle here (from what I understand, there is no assimilation there) ?

3. Eq. (1). Model operator $\mathcal{M}$ not defined

4. L. 228, *to be **statistically independent**,*

5. L. 72, *SGVD $\rightarrow$ SVGD*

6. L. 151, *The first term **in the parenthesis** …*

7. Ll. 254-255, *In algorithm 1 **below**, …*

8. There are occasional useless repetitions. For instance, sentence l. 357 just repeats ll. 346-347.

9. Typos (actually, bugs in word-processing) in ll. 49, 498 and 499 (two typos for the last one).